# The RefinedWeb Dataset for Falcon LLM: Outperforming Curated Corpora with Web Data Only

**The Falcon LLM Team**

**Guilherme Penedo**[2]     **Quentin Malartic**[1]     **Daniel Hesslow**[2]     **Ruxandra Cojocaru**[1]

**Hamza Alobeidli**[1]     **Alessandro Cappelli**[2]     **Baptiste Pannier**[2]

**Ebtesam Almazrouei**[1]     **Julien Launay**[2,3]

Technology Innovation Institute, Abu Dhabi

https://huggingface.co/datasets/tiiuae/falcon-refinedweb

## Abstract

Large language models are commonly trained on a mixture of filtered web data and curated "high-quality" corpora, such as social media conversations, books, or technical papers. This curation process is believed to be necessary to produce performant models with broad zero-shot generalization abilities. However, as larger models requiring pretraining on trillions of tokens are considered, it is unclear how scalable is curation, and whether we will run out of unique high-quality data soon. At variance with previous beliefs, we show that properly filtered and deduplicated web data alone can lead to powerful models; even significantly outperforming models trained on The Pile. Despite extensive filtering, the high-quality data we extract from the web is still plentiful, and we are able to obtain five trillion tokens from CommonCrawl. We publicly release an extract of 600 billion tokens from our REFINEDWEB dataset, and 1.3/7.5B parameters language models trained on it.

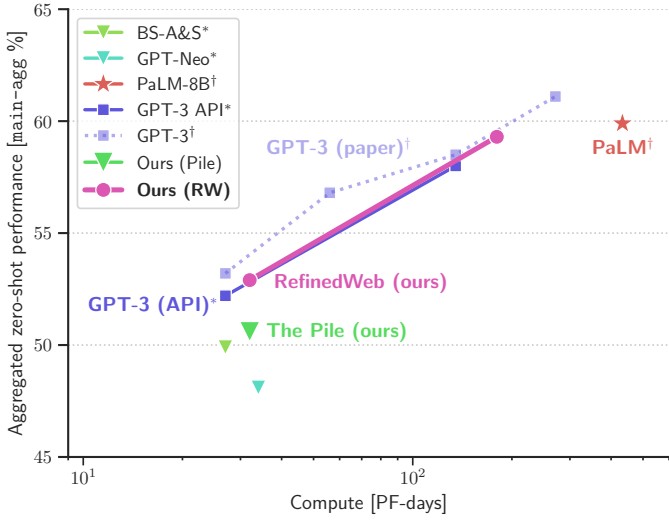

Figure 1: **Models trained on ●REFINEDWEB alone outperform models trained on curated corpora.** Zero-shot performance on our `main-agg` task aggregate (see Section 4.1 for details). At equivalent compute budgets (in PetaFLOPS-days), our models significantly outperform publicly available models trained on ▼ The Pile, and match the performance of the ■ GPT-3 models.

Submitted to the 37th Conference on Neural Information Processing Systems (NeurIPS 2023) Track on Datasets and Benchmarks. Do not distribute.

# 1 Introduction

Progress in natural language processing is increasingly driven by sheer compute scale alone [1]: as more compute is expended to train large language models (LLM), they gain and exhibit powerful emergent capabilities [2, 3]. To best benefit from scaling, recent scaling laws dictate that both model size and dataset size should jointly be increased [4]. This is at variance with earlier findings, which had argued that scaling should focus on model size first and foremost, with minimal data scaling [5].

This joint scaling paradigm raises significant challenges: although plentiful, text data is not infinite, especially so when accounting for data quality and licensing–leading some researchers to argue scaling may soon be bottlenecked by data availability [6]. Concretely, optimally training a GPT-3 sized model (175B parameters) would require no less than 3,500 billion tokens according to [4]. This is twice as much as the largest pretraining datasets publicly demonstrated [4, 7], and ten times more than the largest publicly available English datasets such as OSCAR [8], C4 [9], or The Pile [10].

Massively scaling-up pretraining data is made even more challenging by the fact LLMs are commonly trained using a mixture of web crawls and so-called "high-quality" data [2, 10]. Typical high-quality corpora include curated sources of books, technical documents (e.g., research papers), human-selected web pages, code or social media conversations. The increased diversity and quality brought forth by these curated corpora is believed to be a key component of performant models [11]. Unfortunately, curation is labour intensive: typically, each source requires specialized processing, while yielding a limited amount of data. Furthermore, licensed sources can raise legal challenges.

Nevertheless, most pretraining data is still sourced out of necessity from massive web crawls–as they can be scaled up to trillions of tokens with limited human intervention. However, the quality of this data has traditionally been seen as (much) inferior to that of the manually curated data sources. Even finely processed sources of web data, such as C4 [9] or OSCAR [8], are regarded as inferior to curated corpora for LLMs [12, 11], producing less performant models.

To sustain the ever-increasing needs of larger and larger LLMs, and to streamline data pipelines and reduce the need for human-intensive curation, we explore how web data can be better processed to significantly improve its quality, resulting in models as capable as models trained on curated corpora.

**Contributions.** We make the following contributions:

- We introduce REFINEDWEB, a five trillion tokens web-only English pretraining dataset;
- We demonstrate that **web data alone can result in models outperforming both public and private curated corpora**, challenging current views about data quality;
- **We publicly release a 600B tokens extract of RefinedWeb, and 1/7B parameters LLMs trained on it**, to serve as a new baseline high-quality web dataset for the community.

Table 1: ●REFINEDWEB **improves on existing English pretraining datasets for large language models by combining extensive filtering with stringent deduplication at unprecedented scale.** For additional details, see the full version in Table 12 of Appendix H.3.

| Dataset | Size | Availability | Web | CC Processing | Deduplication |
|---|---|---|---|---|---|
| **MASSIVE WEB DATASETS** | | | | | |
| **C4** | ∼ 360GT | Public | 100% | Rules + NSFW words blocklist | **Exact:** spans of 3 sentences |
| **OSCAR-21.09** | ∼ 370GT | Public | 100% | Built at the line-level | **Exact**: per line (∼ 55% removed) |
| **OSCAR-22.01** | ∼ 283GT | Public | 100% | Line-level rules + optional rules & NSFW URL blocklist | **Exact**: per line (optional, not used for results in this paper) |
| **CURATED DATASETS** | | | | | |
| ■ GPT-3 | 300GT | Private | 60% | Content filter trained on known high-quality sources | **Fuzzy:** MinHash (∼ 10% removed) |
| ▼ The Pile | ∼ 340GT | Public | 18% | jusText for extraction, filter trained on curated data | **Fuzzy**: MinHash (∼ 26% removed) |
| ★ PaLM | 780GT | Private | 27% | Filter trained on HQ data | Unknown |
| **OURS** | | | | | |
| ●REFINEDWEB | ∼ 5,000GT | Public (500GT) | 100% | trafilatura for text extraction, document and line-level rules, NSFW URL blocklist | **Exact & fuzzy**: exact substring+MinHash (∼ 50% removed) |

## 2 Related works

**Pretraining data for large language models.** Both GPT and BERT identified the importance of datasets with long, coherent documents [13, 14]. Moving from sentence-wise datasets [15], they instead leveraged document-focused, single-domain corpora like Wikipedia or BookCorpus [16]. As models increased in scale, datasets based on massive web-scrape gained prevalence [8, 9]. However, further work argued that these untargeted web scrape fell short of human-curated data [17], leading to the wide adoption of curated datasets such as The Pile [10], combining web data with books, research articles, conversations, and more. At scale, it has been proposed to emulate the human curation process by leveraging weak signals: for instance, by crawling the top links of a forum [18]. Targeted corpora can also produce domain-specific models [19], or broaden the expressiveness of models (e.g., for conversational modalities [20, 21]). Latest large language models [2, 12, 22, 23] are trained on giant aggregated corpora, combining both massive web-scrape and so-called "high-quality" curated single-domain sources. These targeted sources are often upsampled–from one to five times is most common–to increase their representation in the final dataset. The assumed diversity and higher-quality brought fourth by these aggregated datasets is thought to be central to model quality; web data alone is considered insufficient to train powerful large language models [24, 11].

**Pipelines for web data.** Massive web datasets are typically built upon CommonCrawl, a publicly available scrape of the internet. Working with data scraped from all over the internet presents unique challenges: notably, a significant portion is machine-generated spam or pornographic content [25, 26]. Accordingly, training on unfiltered web data is undesirable, resulting in poorly performing models [9]. Modern pipelines focus on filtering out undesirable content [27]. Broadly speaking, these pipelines usually combine a variety of stages: (1) *language identification*, leveraging inexpensive n-gram models (e.g., fastText [28]); (2) *filtering rules and heuristics*, such as only keeping lines with valid punctuation, discarding lines with too many symbols, or removing documents containing banned words [29, 9]; (3) *ML-based quality filtering*, using lightweight models trained on known gold data to identify similar high-quality web documents [27, 2]; (4) *deduplication*, removing either exact duplicate spans or similar documents [30]. While some filtering is necessary, excessive filtering can introduce undesirable biases: this can overly impact minorities [31], motivating the adoption of practices such as pseudo-crawling, wherein allowed URLs are manually curated [32].

**Deduplication.** Deduplication removes repeated extracts and documents from a dataset: these could either be exact matches, identical in every character, or approximate matches, based on some similarity metric. For exact duplicates, it is common to match exact substrings of a minimum length using suffix arrays [33]. For fuzzy duplicates, methods based on locally-sensitive hashes such as MinHash [34] or SimHash [35] have seen wide adoption [2, 36, 12]. Recently, [37] has proposed to leverage embeddings to imbue semantic understanding in approximate matching algorithms. Deduplication has been identified as playing a significant role in improving language models [38, 30]. Notably, it reduces memorization [39], which is especially problematic in large models [40]. Furthermore, repeated data has been shown to be increasingly harmful to model quality as parameter count increases [41]: for a 1B parameters model, a hundred duplicates are harmful; at 175B, even a few duplicates could have a disproportionate effect. Concurrently to this work, the Pythia suite of models found that deduplicating The Pile had a limited impact on zero-shot performance [42], questioning whether deduplication is as relevant for curated corpora as it for predominantly web-based datasets as studied in Lee et al. [30].

We provide an overview of some widely adopted pretraining English datasets for LLMs in Table 1, with additional information in Table 12 of Appendix H.3. We also note that recent popular open models [43, 7] often indirectly leverage The Pile [10] by doing a mix-and-match of its components.

With REFINEDWEB, we extend upon the state-of-the-art in three ways: (1) we aggregate and combine best-practices for document preparation and filtering across multiple pipelines, and introduce line-wise corrections to fix lingering issues with text extraction; (2) we combine both exact and fuzzy deduplication at very large-scale; (3) the scale of our final dataset is unique, with a total 5,000 billion tokens, and a 600 billion tokens extract available for public use with permissive licensing. Training large models on RefinedWeb also lead us to challenge the commonly held belief that web data is worse than curated corpora, as our models outperform others trained on so-called "high-quality" data.

# 3 Macrodata Refinement and RefinedWeb

We introduce **MDR** (MacroData Refinement), a pipeline for filtering and deduplicating web data from CommonCrawl at very large scale. Using MDR, we produce REFINEDWEB, an English pretraining dataset of five trillion tokens based on web data only. We leverage strict filtering and stringent deduplication to uplift the quality of web data, distilling it down to a corpus matching the quality of aggregated corpora used to train state-of-the-art models.

**Design principles.** We abide by the following guidelines:

- **Scale first.** We intend MDR to produce datasets to be used to train 40-200B parameters models, thus requiring trillions of tokens [4]. For English-only RefinedWeb, we target a size of 3-6 trillion tokens. Specifically, we eschew any labour intensive human curation process, and focus on CommonCrawl instead of disparate single-domain sources.
- **Strict deduplication.** Inspired by Lee et al. [30], which demonstrated the value of deduplication for LLMs, we implement a rigorous deduplication pipeline. We combine both exact and fuzzy deduplication, and use strict settings leading to high removal rates.
- **Neutral filtering.** To avoid introducing further undesirable biases into the model [31, 44], we avoid using ML-based filtering outside of language identification. We stick to simple rules and heuristics, and use only URL filtering for adult content.

## 3.1 Document preparation: reading data, filtering URLs, extracting text, and language identification

**Reading the data.** CommonCrawl is available in either WARC (raw HTML response), or WET files (preprocessed to only include plain text). Individual files correspond to a page/document/sample at a given URL. WET files would spare us from running our own HTML extraction; however, in line with previous works [10, 12], we found WET files to include undesirable navigation menus, ads, and other irrelevant texts. Accordingly, we start from raw WARC files, read with the `warcio` library.

**URL filtering.** Before undertaking any compute-heavy processing, we perform a first filtering based on the URL alone. This targets fraudulent and/or adult websites (e.g., predominantly pornographic, violent, related to gambling, etc.). We base our filtering on two rules: (1) an aggregated blocklist of 4.6M domains; (2) a URL score, based on the presence of words from a list we curated and weighed by severity. We found that commonly used blocklists include many false positives, such as popular blogging platforms or even pop culture websites. Furthermore, word-based rules (like the one used in

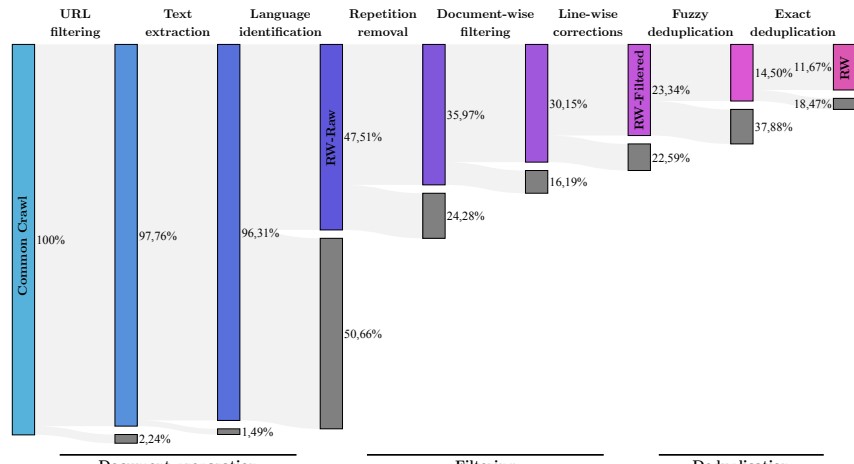

Figure 2: **Subsequent stages of Macrodata Refinement remove nearly 90% of the documents originally in CommonCrawl.** Notably, filtering and deduplication each result in a halving of the data available: around 50% of documents are discarded for not being English, 24% of remaining for being of insufficient quality, and 12% for being duplicates. We report removal rate (grey) with respect to each previous stage, and kept rate (shade) overall.

Table 2: **Macrodata Refinement aggregates best practices from the state-of-the-art and novel approaches (URL scoring, line-wise filtering, etc.) to produce high-quality web data.** On deduplication, we note that MDR is unique in both the scale at which it is performed, and in applying subsequently fuzzy and exact substring methods to improve coverage and scalability.

| DOCUMENT PREPARATION | | | FILTERING | | DEDUPLICATION | |
|---|---|---|---|---|---|---|
| **URL filtering** | **Text extraction** | **Language identification** | **Document-wise filtering** | **Line-wise filtering** | **Deduplication** | **URL deduplication** |
| Aggregated blocklist, URL scoring, common HQ sources blocked | From WARC using `warcio`, `trafilatura` for extraction | `fastText` classifier from CCNet, thresholding on top language score | In-document repetition removal and quality heuristics from MassiveWeb | Remove undesirable lines (call to actions, navigation buttons, social counters, etc.) | Fuzzy deduplication w/ MinHash + exact substring deduplication w/ suffix arrays | Remove URLs revisited across Common-Crawl dumps |
| Appendix I.1 | Barbaresi [46] | Wenzek et al. [27] | Rae et al. [12] | Appendix I.2 | Lee et al. [30] | Section 3.3 |

C4, [9]) can easily result in medical and legal pages being blocked. Our final detailed rules based on this investigation are shared in Appendix I.1. Since we intend RefinedWeb to be used as part of an aggregate dataset along with curated corpora, we also filtered common sources of high-quality data: Wikipedia, arXiv, etc. The detailed list is available in Appendix I.1.3.

**Text extraction.**    We want to extract only the main content of the page, ignoring menus, headers, footers, and ads among others: Lopukhin [45] found that `trafilatura` [46] was the best non-commercial library for retrieving content from blog posts and news articles. Although this is only a narrow subset of the kind of pages making up CommonCrawl, we found this finding to hold more broadly. We use `trafilatura` for text extraction, and apply extra formatting via regular expressions: we limit new lines to two consecutive ones, and remove all URLs.

**Language identification.**    We use the fastText language classifier of CCNet [27] at the document-level: it uses characters n-gram and was trained on Wikipedia, supporting 176 languages. We remove documents for which the top language scores below 0.65: this usually corresponds to pages without any natural text. For this paper, we focus on English; RefinedWeb can also be derived for other languages, see Appendix F for details.
The data we retrieve at this stage, called **RW-RAW**, corresponds to what we can extract with the minimal amount of filtering. At this stage, only 48% of the original documents are left, mostly filtered out by language identification (and a small fraction by failures of the text extraction).

### 3.2   Filtering: document-wise and line-wise

**Repetition removal.**    Due to crawling errors and low-quality sources, many documents contain repeated sequences: this may cause pathological behavior dowstream [47]. The later deduplication stage could catch this, but it is cheaper to catch it earlier document-wise. We implement the heuristics of Rae et al. [12], removing any document with excessive line, paragraph, or n-gram repetitions.

**Document-wise filtering.**    A significant fraction of pages are machine-generated spam, made predominantly of lists of keywords, boilerplate, or sequences of special characters. Such documents are not suitable for language modeling; to filter them out, we adopt the quality filtering heuristics of Rae et al. [12]. These remove outliers in terms of overall length, symbol-to-word ratio, and other criteria ensuring the document is natural language. We note we adapted these filters on a per language basis, as they may result in overfiltering if naively transferred from English to other languages.

**Line-wise corrections.**    Despite the improvements brought forth by using `trafilatura` instead of relying on preprocessed files, many documents remain interlaced with undesirable lines (e.g., social media counters [3 `comments`], navigation buttons [`Home`]). Accordingly, we devised a line-correction filter, targeting these undesirable items leftover from text extraction imperfections. If these corrections remove more than 5% of a document, we remove it entirely. See Appendix I.2 for details.

The data we retrieve at this stage has gone through all of the filtering heuristics in the MDR pipeline. We refer to this dataset as **RW-FILTERED**. Only 23% of the documents of CommonCrawl are left, with around 50% of the documents of RW-Raw removed by the filtering.

### 3.3 Deduplication: fuzzy, exact, and across dumps

After filtering, although data quality has improved, a large fraction of the content is repeated across documents. This may be due to the crawler indirectly hitting the same page multiple times, to boilerplate content being repeated (e.g., licences), or even to plagiarism. These duplicates can strongly impact models, favoring memorization instead of generalization [30, 41]. Since deduplication is expensive, it has seen limited adoption in public datasets [8, 9]. We adopt an aggressive deduplication strategy, combining both fuzzy document matches and exact sequences removal.

**Fuzzy deduplication.** We remove similar documents by applying MinHash [34]: for each document, we compute a sketch and measure its approximate similarity with other documents, eventually removing pairs with high overlap. MinHash excels at finding templated documents: licenses with only specific entities differing, placeholder SEO text repeated across websites–see examples of the biggest clusters in Appendix J.1. We perform MinHash deduplication using 9,000 hashes per document, calculated over 5-grams and divided into 20 buckets of 450 hashes. We found that using less aggressive settings, such as the 10 hashes of The Pile [10], resulted in lower deduplication rates and worsened model performance. See Appendix I.3.1 for more details about our MinHash setup.

**Exact deduplication.** Exact substring operates at the sequence-level instead of the document-level, finding matches between strings that are exact token-by-token matches by using a suffix array [33] (e.g., specific disclaimers or notices, which may not compromise the entire document as showcased in Appendix J.2). We remove any match of more than 50 consecutive tokens, using the implementation of Lee et al. [30]. We note that exact substring alters documents, by removing specific spans: we also experimented with dropping entire documents or loss-masking the duplicated strings instead of cutting them, but this didn't result in significant changes in zero-shot performance–see Appendix I.3.2.

**URL deduplication.** Because of computational constraints, it is impossible for us to perform deduplication directly on RW-Filtered. Instead, we split CommonCrawl into 100 parts, where each part contains a hundredth of each dump, and perform deduplication on individual parts. Most of the larger duplicate clusters (e.g., licences, common spams) will be shared across parts, and effectively removed. However, we found that CommonCrawl dumps had significant overlap, with URLs being revisited across dumps despite no change in content. Accordingly, we keep a list of the URLs of all samples we have kept from each part, and remove them from subsequent parts being processed.

Table 3: **To evaluate models trained on RefinedWeb and compare to the state-of-the-art, we build four aggregates across 18 tasks on which to measure zero-shot performance.** small was built for internal ablations, based on tasks with consistent performance at small scale, core is based on tasks commonly reported for public suites of models [48, 42], main is based on tasks from the GPT-3 and PaLM paper [2, 22], and ext is based on tasks used by the BigScience Architecture and Scaling group [11]. We flag with † results obtained in an arbitrary evaluation setup, and with ∗ results obtained with the EAI Harness [49], which we also employ for all our models.

| Tasks | Type | Random | small | core | main | ext |
|---|---|---|---|---|---|---|
| HellaSwag [50] | Sentence completion | 25.0 | ✓ | ✓ | ✓ | ✓ |
| LAMBADA [51] | Sentence completion | 0.0 | | ✓ | ✓ | ✓ |
| Winogrande [52] | Coreference resolution | 50.0 | ✓ | ✓ | ✓ | ✓ |
| PIQA [53] | Multiple-choice question answering | 50.0 | ✓ | ✓ | ✓ | ✓ |
| ARC [54] | Natural language inference | 25.0 | ✓ | ✓ | ✓ | ✓ |
| OpenBookQA [55] | Multiple-choice question answering | 25.0 | | ✓ | ✓ | ✓ |
| BoolQ [56] | Multiple-choice question answering | 50.0 | ✓ | | ✓ | ✓ |
| COPA [57] | Sentence completion | 50.0 | | | ✓ | ✓ |
| CB [58] | Natural language inference | 33.3 | | | ✓ | ✓ |
| RTE [59] | Natural language inference | 50.0 | | | ✓ | ✓ |
| ReCoRD [60] | Question answering | 0.0 | | | ✓ | |
| ANLI [61] | Natural language inference | 33.3 | | | ✓ | |
| LogiQA [62] | Multiple-choice question answering | 25.0 | | | | ✓ |
| HeadQA [63] | Multiple-choice question answering | 20.0 | | | | ✓ |
| MathQA [64] | Multiple-choice question answering | 20.0 | | | | ✓ |
| PROST [65] | Paraphrase identification | 50.0 | | | | ✓ |
| PubMedQA [66] | Multiple-choice question answering | 50.0 | | | | ✓ |
| SciQ [67] | Multiple-choice question answering | 25.0 | ✓ | | | ✓ |

Table 4: **Curation is not a silver bullet for zero-shot generalization: small-scale models trained on ●REFINEDWEB outperform models trained on web data (C4, OSCAR), and on curated corpora (▼ The Pile).** Average accuracy in zero-shot on the `small-agg` aggregate. All models trained with identical architectures and pretraining hyperparameters, for the same amount of tokens. We find that OSCAR-22.01 underperforms other datasets signficantly, perhaps because deduplication is only optional. C4 is a strong baseline, with OSCAR-21.09 lagging slightly behind, but we find that RefinedWeb outperforms both web datasets and the most popular curated dataset, The Pile. Both filtering and deduplication contribute significantly to improving zero-shot performance.

| | MASSIVE WEB DATASETS | | | CURATED | OURS | | |
|---|---|---|---|---|---|---|---|
| | OSCAR-21.09 | OSCAR-22.01 | C4 | ▼ The Pile | RW-Raw | RW-Filtered | ●REFINEDWEB |
| **1B@27GT** | 55.0% | 52.7% | 55.7% | 53.4% | 52.7% | 54.3% | **56.2%** |
| **3B@60GT** | 59.1% | 55.9% | 59.6% | 57.9% | 57.4% | 58.2% | **59.8%** |

## 4 Experiments

We now validate that models trained on RefinedWeb can match the zero-shot performance obtained with curated corpora and by state-of-the-art models. We first discuss our evaluation and pretraining setup, and models with which we compare. We perform experiments at small scale to internally compare with other datasets, and ablate the stages of RefinedWeb (raw, filtered, final). Then, we scale to 1B and 7B models trained on 350GT to compare with the state-of-the-art. Finally, we apply the MDR pipeline to existing datasets, and show that it can potentially deliver further improvements.

### 4.1 Setting

**Evaluation.** At variance with previous works studying pretraining datasets [12, 30], we focus our evaluation on zero-shot generalization across many tasks rather than measuring validation loss. Perplexity alone can be at odds with end-task performance [68], and modern works on LLMs predominantly report zero-shot performance [2, 12, 22]. Furthermore, zero-shot generalization is the "natural" setting for autoregressive decoder-only models, in which they perform best [69]. Our evaluation setup is inspired by the one used by the architecture and scaling group of Big Science [11].

We base our evaluation on the Eleuther AI evaluation harness [49], allowing us to evaluate across a wide range of tasks. We identified aggregates allowing us to: (1) obtain signal (i.e., non zero zero-shot performance) at small scale for ablations; (2) compare with results reported by other models. We outline these aggregates `small` (for ablations), and `core`, `main`, `ext` (for comparisons) in Table 3.

Comparisons across models trained and evaluated in different settings are difficult to untangle, as many externalities may influence the results (e.g., numerical precision of training vs inference, prompts used). We distinguish three levels of comparisons: (1) internal comparisons, with models trained and evaluated within our codebase, for which only the pretraining datasets differ; (2) benchmark-level comparisons, with models trained with a different codebase but evaluated with the Eleuther AI harness, taking results from [11, 70, 71, 48], thereafter flagged with a ∗; (3) external comparisons with [2, 22], thereafter flagged with a †. For further details on evaluation, see Appendix H.1.

**Models.** We train 1B, 3B, and 7B parameters autoregressive decoder-only models, based on configurations and hyperparameters similar to GPT-3 [2], diverging mostly on our use of ALiBi [72]. We use FlashAttention [73] in a custom codebase. We train internal models on both The Pile and RefinedWeb to control for deviations caused by our pretraining setup–we found The Pile models to perform in-line with others. For small-scale and ablation studies (first half of Section 4.2; Section 4.3), we train models to optimality according to the scaling laws of Hoffmann et al. [4]: on 27B and 60B tokens respectively for our 1B and 3B parameters models. For the main experiments demonstrating our approach (Falcon-RW models in Section 4.2), we train the models to 350GT, in line with popular public models [2, 74, 23]. Note that we do not compare against the recently introduced LLaMA models [7], as the smallest of them is trained on x2.5 more compute than our largest model, preventing a meaningful comparison from being made dataset-wise. For a more in-depth overview of the models and pretraining datasets with which we compare, see Appendix H.

## 4.2 Can web data alone outperform curated corpora?

We endeavour to demonstrate that web data alone can result in models outperforming models trained on curated corpora. To do so, we first perform a small-scale study with 1B and 3B parameters models trained to optimality (27GT and 60GT) on popular web and curated datasets. Then, we scale up to 1B and 7B models trained on 350GT, and compare zero-shot generalization to state-of-the-art models.

**Small-scale study.** We first consider public web datasets (OSCAR-2019 [8], OSCAR-2022 [75], C4 [9]), The Pile [10] as the most popular publicly available curated dataset, and variations of RefinedWeb (RW-Raw, RW-Filtered, and RW as described in Section 3). All models are trained with the same architecture, for the same amount of tokens, and using the same internal codebase; they are also all evaluated within the same framework–only pretraining datasets differ.

Results averaged on the `small` aggregate of 6 tasks are presented in Table 4. We observe relatively strong performance of all web datasets compared to The Pile, showcasing that curation is not a silver bullet for performant language models. We find C4 to be a strong pretraining dataset, in line with the findings of Scao et al. [11]–however, The Pile underperforms more in our benchmarks. The disappointing results on OSCAR-22.01 may be due to the dataset being distributed without deduplication by default. Regarding RefinedWeb, both filtering and deduplication significantly improve performance. We also note that a 3B@60GT model trained on OSCAR-22.1 performs *worse* than a 1B@27GT model trained on RefinedWeb: data alone accounts for a 4x difference in pretraining compute, highlighting that compute budgets alone cannot compensate efficiently for inadequate data.

**Full-scale models.** We now validate these results with comparisons with state-of-the-art models. We scale our previous experiments by training 1B and 7B models on 350GT; we also train a 1B model on 350GT on The Pile, as a control for the influence of our pretraining setup. We compare with the following models: the GPT-3 series [2], the FairSeq series [76], the GPT-Neo(X)/J models [77, 74, 70], the OPT series [43], the BigScience Architecture and Scaling Pile model [11], PaLM-8B [22], Aleph Alpha Luminous 13B [71], the Pythia series [42], and the Cerebras-GPT series [48]. For GPT-3, we distinguish between results obtained through the API (`babbage` and `curie`) with the the EleutherAI LM evaluation harness [49] (*), and results reported in their paper, with a different evaluation setup (†). For PaLM and OPT, results were obtained also with a different evaluation suite (†); for most other models they were obtained with the evaluation harness (*), allowing for more direct comparisons.

Results on `main-agg` are presented in Figure 1, and in Figure 3 for `core-agg` and `ext-agg`. We find that open models consistently underperform models trained on private curated corpora, such

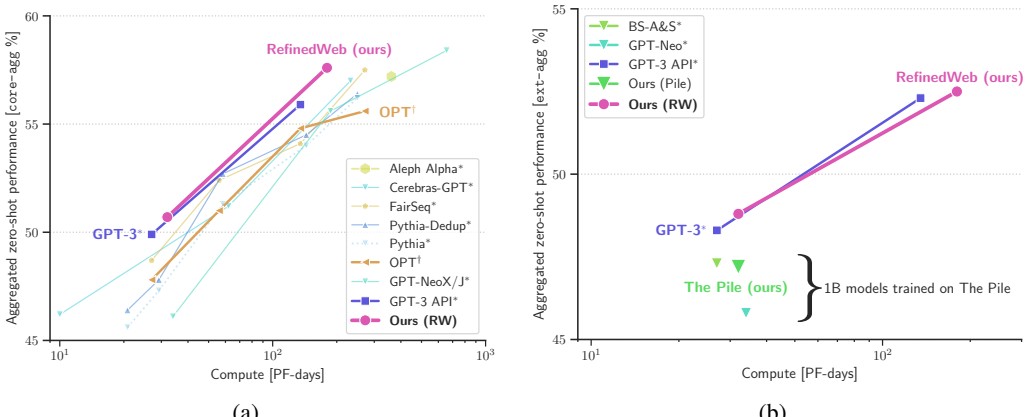

(a)  (b)

Figure 3: **Models trained on ●REFINEDWEB alone outperform models trained on curated corpora.** Zero-shot performance averaged on our `core-agg` (left) and `ext-agg` (right) task aggregates (see Section 4.1 for details, and Figure 1 for results on `main-agg`). Existing open models fail to match the performance of the original GPT-3 series (left); however, models trained on RefinedWeb significantly outperform models trained on ▼ The Pile: including our direct comparison model (right), ruling out our pretraining setup as the main source of increased performance. In fact, our RefinedWeb models even match the performance of the ■ GPT-3 models.

Table 5: **Although improvements from filtering are not systematic across datasets, deduplication brings a steady performance boost across the board.** Zero-shot accuracy averaged on `small-agg` aggregate; [+x.x] reports absolute gains compared to base, removal rates reported against base. Due to limitations in our pipeline, we cannot apply the deduplication stage independently for RefinedWeb.

| | MASSIVE WEB DATASETS | | | CURATED | OURS |
|---|---|---|---|---|---|
| | OSCAR-21.09 | OSCAR-22.01 | C4 | ▼ Pile | ●RefinedWeb |
| **Base** | 55.0% | 52.7% | **55.7%** | 53.4% | 52.7% |
| **Filtered** | 55.4% [+.4] | 52.3% [-.4] | **56.2%** [+.5] | 54.2% [+.8] | 54.3% [+1.6] |
| *removal rate* | *-25.0%* | *-39.8%* | *-16.4%* | *-27.1%* | *-50.8%* |
| **Deduplicated** | 55.6% [+.6] | 55.6% [+2.9] | **55.9%** [+.2] | 54.5% [+1.1] | |
| *removal rate* | *-10.8%* | *-60.8%* | *-7.59%* | *-45.3%* | |
| **Filt.+Dedup.** | 55.5% [+.5] | 55.4% [+2.7] | **56.4%** [+.7] | 55.2% [+1.8] | 56.2% [+3.5] |
| *removal rate* | *-28.2%* | *-62.2%* | *-17.9%* | *-66.0%* | *-75.4%* |

264 as GPT-3–even when using a similar evaluation setup. Conversely, models trained on RefinedWeb
265 are able to match the performance of the GPT-3 series using web data alone, even though common
266 high-quality sources used in The Pile are excluded from RefinedWeb (see Table 14 in Appendix).
267 Finally, we note that our internal model trained on The Pile performs in line with the BigScience
268 Architecture and Scaling model; this highlights that our pretraining setup is unlikely to be the main
269 source of increased performance for models trained on RefinedWeb.

> **Finding.** Challenging beliefs on data quality, filtered and deduplicated web data *alone* allows models to match the natural language tasks performance of models trained on curated data.

## 4.3 Do other corpora benefit from MDR?

271 Ablating the contributions and evaluating the performance of individual components in the MDR
272 pipeline is difficult: for most heuristics, there is no agreed-upon ground truth, and changes may be
273 too insignificant to result in sufficient zero-shot signal after pretraining. In the first half of Section 4.2,
274 we identified that subsequent stages of RefinedWeb (raw, filtered, final) led to improvements in
275 performance. In this section, we propose to apply independently the filtering and deduplication stages
276 of MDR to popular pretraining datasets, studying whether they generalize widely.

277 We report results on the `small-agg` in Table 5. First, we find that improvements from filtering
278 are not systematic. On The Pile, we had to adjust our line length and characters ratio heuristics to
279 avoid expunging books and code. Despite improvements on OSCAR-21.09, C4, and The Pile, our
280 filters worsen performance on OSCAR-22.01; generally, removal rates from filtering are not strongly
281 correlated with downstream accuracy. Conversely, deduplication delivers a steady boost across all
282 datasets, and removal rates are better correlated with zero-shot improvements. OSCAR-21.09 and
283 C4 are already well deduplicated, while The Pile and OSCAR-22.01 exhibit 40-60% duplicates.
284 OSCAR-22.01 is distributed without deduplication by default; for The Pile, this is consistent with
285 the findings of Zhang et al. [43]. Finally, combining filtering and deduplication results in further
286 improvements; although performance is now more uniform across datasets, differences remain,
287 suggesting that flaws in the original text extraction and processing are not fully compensated for.

288 By processing C4 with MDR, we are able to obtain subsets of data which might slightly outperform
289 RefinedWeb; this combines both the stringent filtering of C4 (e.g., strict NSFW word blocklist,
290 3-sentence span deduplication) with our own filters and deduplication. While this results in rejection
291 rates that are unacceptable for our target of 3-6 trillions tokens, this is an interesting perspective for
292 shorter runs, which may be able to extract extremely high-quality subsets from large datasets.

> **Finding.** While filtering heuristics may require source-dependent tuning, stringent deduplication improves zero-shot performance across datasets consistently.

## 5   Limitations

**Biases and harmfulness.**   We conduct an analysis of the toxicity of RefinedWeb in Figure 5 of the Appendix. We find RefinedWeb to be about as toxic as The Pile, based on the definition of toxicity of the Perspective API: "content that is rude or disrespectful". Notably, this definition does not cover social biases or harmfulness. Although it is unlikely that our pipeline introduces further issues than is already documented for popular datasets, we encourage quantitative work on our public extract.

**Performance beyond natural language.**   Our evaluation aggregates are overwelmingly targeting natural language tasks, and do not include code or mathematics evaluation–which are popular use cases for fully-fledged models. A natural question may be whether web data alone is sufficient to achieve strong code/mathematics performance; we do not think this is the case, and encourage practionners to combine RefinedWeb with code datasets such as The Stack [78] when training modles. However many of our findings apply equally: notably, Li et al. [79] found that deduplication helped with code data collected from GitHub as well. Broadly speaking, like web data is massively collected from CommonCrawl, code data is usually collected from GitHub, before undergoing extensive filtering and deduplication. This is similar to the spirit of RefinedWeb, and does not rely on a collection of curated sources. Finally, we note that specific domains (e.g., code, technical papers) exist on a spectrum, and that general natural language improvements may benefit technical tasks too: for instance, we find that models trained on RefinedWeb outperform on PubMedQA models trained on The Pile, despite not including any explicit medical data (The Pile includes PubMed).

**And beyond pretraining...**   Our study is strictly limited to language model pretraining, and does not address finetuning existing models. We note the value of high-quality samples for downstream specialization, for instance for improving chattiness or instruction-following capabilities [80].

**Multiple epochs.**   Instead of looking for "unique" tokens for a trillion-scale pretraining dataset, one could simply repeat data over multiple epochs. Popular models like OPT and NeoX-20B train on up to 2 epochs [43, 70], and most curated datasets upsample corpora 2-5 times [2, 10]. However, Hernandez et al. [41] has recently shown that models with 100B+ parameters may be sensitive to even just a few epochs. Orthogonal to our work one could explore tradeoffs in the data-constrained regime: can deduplication help sustain more epochs? Are multiple epochs on higher quality data better than one epoch on lower quality data? See Appendix G.3 for a more in-depth discussion.

**Other results on deduplication.**   Biderman et al. [42] found a limited impact on zero-shot performance from deduplicating The Pile; we discuss in Appendix H.2 and suspect deduplication may be unreasonably effective on web datasets because it predominantly removes low quality content (see Appendix J for top samples). Muennighoff et al. [81] studied scaling laws for multiple epochs, and found that up to four epochs carried limited degradation–however, we note that many of the duplicates we find are present hundred to thousands of time in the raw data, far from this safe regime.

## 6   Conclusion

As LLMs are widely adopted, models trained past the recommendations of scaling laws are bound to become increasingly common to amortize inference costs [7]. This will further drive the need for pretraining datasets with trillions of tokens, an order of magnitude beyond publicly available corpora. We have demonstrated that stringent filtering and deduplication could result in a five trillion tokens web only dataset suitable to produce competitive models, even outperforming LLMs trained on curated corpora. We publicly release a 600GT extract of RefinedWeb, and note that RefinedWeb has already been used to train state-of-the-art language models, such as Falcon-40B [82].

We publicly release the following artefacts:

- **A 600B tokens extract of RefinedWeb:** `https://huggingface.co/datasets/tiiuae/falcon-refinedweb`;
- The 1B and 7B models trained on RefinedWeb in this paper: `https://huggingface.co/tiiuae/falcon-rw-1b` and `https://huggingface.co/tiiuae/falcon-rw-7b`.

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
