# B  NeurIPS Datasets and Benchmarks track details

## B.1  Accessing and using the data

To host artefacts related to this paper, we are leveraging the HuggingFace Hub. This both guarantees long-term availability, and standardization allowing for interoperability with tools built by the community. Our 600B tokens public RefinedWeb extract is made available using the `datasets` library [83], and our 1B and 7B models are released with the `transformers` library [84].

- **Falcon-RefinedWeb**, the 600B tokens extract we make publicly available: `https://huggingface.co/datasets/tiiuae/falcon-refinedweb`, licensed under an ODC-By 1.0 license, and users should also abide to the CommonCrawl ToU;
- **Falcon-RW-1/7B**, the two models we have pretrained on RefinedWeb-only for this paper: `https://huggingface.co/tiiuae/falcon-rw-1b` and `https://huggingface.co/tiiuae/falcon-rw-7b`, both licensed under an Apache 2.0 license.

The complete model and data cards on the hub contain useful code examples for getting started with our public released assets. For the cards included in this supplementary, we focused on documentation rather than technical details.

The DOI associated with the public RefinedWeb extract is `10.57967/hf/0737`.

## B.2  Compute ressources

We used resources from AWS Sagemaker, training on P4d instances with eight A100 40GB per node. Nodes were interconnected with 50Gb/s of EFA interconnect. We estimate to have used 55,000A100-hour for this project: 35,000 for the 7B model; 5,000 for the 1B; and 15,000 for small scale ablations and earlier experiments not reported in this paper. See Appendix I.4 for details on CPU processing.

## B.3  Statement on resubmission

This work was previously submitted at ICML 2023, and was rejected with scores 8/7/6/3. Compared to this earlier version, we have made the following improvements based on reviewers' feedback:

- **Added further comparisons with state-of-the-art models**, such as OPT and Pythia.
- **Analysed the toxicity of RefinedWeb**, as presented in Figure 5, leveraging the Perspective API to demonstrate the prevalence of toxic content in RefinedWeb to be similar to The Pile.
- **Added an ablation on the effect of our pipeline on other web datasets**, as presented in Appendix G.1, demonstrating its wide applicability and generalizing our findings on the value of filtering and deduplication.
- **Improved the overall presentation**, by surfacing content from the Appendix on our evaluation and on the limitations of our work.

## B.4  Statement on license

We release the dataset under the ODC-By-1.0 license, further requesting from users that they abide by the CommonCrawl Terms of Use. This is inspired by other public massive web crawls, such as C4 [9]. This licensing permits sharing, reuse, and adaptation of the dataset, for any purpose, as long as

both CommonCrawl and this paper are acknowledged and provided appropriate credit. The purpose
of this statement is to clarify the responsibilities and liabilities associated with the use of this dataset.
While we have made every effort to ensure the accuracy and legality of the data contained within this
dataset, we cannot guarantee its absolute completeness or correctness due to its scale.

Therefore, in the event that any rights, legal or otherwise, are violated through the use of this
dataset, including but not limited to copyright infringement, privacy violations, or misuse of sensitive
information, we, the authors, assume no liability for such violations.

By utilizing this dataset, you agree that any consequences, legal or otherwise, arising from the use of
this dataset will be the sole responsibility of the user. You acknowledge that you will exercise due
diligence and adhere to all applicable laws, regulations, and ethical guidelines when using the dataset.

By accessing, downloading, or using this dataset, you signify your acceptance of this statement and
your commitment to abide by the terms and conditions of the ODC-By-1.0 license and by Terms of
Use of CommonCrawl.

If you do not agree with the terms of this statement, the ODC-By-1.0 license, or the CommonCrawl
Terms of Use, you are not authorized to use this dataset.

 # C  RefinedWeb Datasheet

| MOTIVATION | |
|---|---|
| **For what purpose was the dataset created?** | RefinedWeb was created to serve as a large-scale dataset for the pretraining of large language models. It may be used on its own, or augmented with curated sources (e.g., Wikipedia, StackOverflow). |
| **Who created the dataset and on behalf of which entity?** | The dataset was created by the Technology Innovation Institute. |
| **Who funded the creation of the dataset?** | The creation of the dataset was privately funded by the Technology Innovation Institute. |
| **Any other comment?** | RefinedWeb is built on-top of CommonCrawl, using the Macro-data Refinement Pipeline, which combines content extraction, filtering heuristics, and deduplication. In designing RefinedWeb, we abided to the following philosophy: (1) **Scale first.** We intend MDR to produce datasets to be used to train 40-200B parameters models, thus requiring trillions of tokens [4]. For English-only RefinedWeb, we target a size of 3-6 trillion tokens. Specifically, we eschew any labour intensive human curation process, and focus on CommonCrawl instead of disparate single-domain sources. (2) **Strict deduplication.** Inspired by the work of [30], which demonstrated the value of deduplication for large language models, we implement a rigorous deduplication pipeline. We combine both exact and fuzzy deduplication, and use strict settings leading to removal rates far higher than others have reported. (3) **Neutral filtering.** To avoid introducing further undesirable biases into the model [31, 44], we avoid using ML-based filtering outside of language identification. We stick to simple rules and heuristics, and use only URL filtering for adult content. |
| COMPOSITION | |
| **What do the instances that comprise the dataset represent?** | Instances are text-only documents, corresponding to single web pages. |
| **How many instances are there in total?** | RefinedWeb contains ∼10 billion documents, or around 5 trillion tokens. The public version is a subset representing a tenth of the full version. |
| **Does the dataset contain all possible instances or is it a sample (not necessarily random) of instances from a larger set?** | RefinedWeb is built using all CommonCrawl dumps until the 2023-06 one; it could be updated with additional dumps as they are released. The public release of RefinedWeb is a 600GT random extract of the 5,000GT of the full dataset (limited to 600GT for commercial reasons). For experiments, we randomly sampled from the public extract, or earlier development versions. |
| **What data does each instance consist of?** | Each instance is a text-only document, with metadata about its origin in CommonCrawl and source page URL. We also distribute a multimodal version of RefinedWeb, containing interlaced links to images. |
| **Is there a label or target associated with each instance?** | No. |
| **Is any information missing from individual instances?** | No. |
| **Are relationships between individual instances made explicit?** | No. |

| | |
|---|---|
| **Are there recommended data splits?** | No. |
| **Are there any errors, sources of noise, or redundancies in the dataset?** | Despite our best efforts to filter content that does not qualify as natural language, and to deduplicate documents, our pipeline may let through documents that may be considered as errors or redundant. |
| **Is the dataset self-contained, or does it link to or otherwise rely on external resources?** | The base version of the dataset is self-contained, but the multimodal version is interlaced with links to images–these are not distributed as part of the dataset, and constitute an external source. |
| **Does the dataset contain data that might be considered confidential?** | All documents in RefinedWeb have been publicly available online. |
| **Does the dataset contain data that, if viewed directly, might be offensive, insulting, threatening, or might otherwise cause anxiety?** | Yes, as this type of data is prevalent on the internet, it is likely our dataset contains such content. Notably, we estimate the prevalence of toxic content in the dataset to be similar to The Pile (Figure 5). |



**COLLECTION**



| | |
|---|---|
| **How was the data associated with each instance acquired?** | We downloaded with `warcio` publicly available .WET files from the CommonCrawl foundation. |
| **What mechanisms or procedures were used to collect the data?** | We refer to the CommonCrawl website (`commoncrawl.org`) for details on how they collect data. |
| **If the dataset is a sample from a larger set, what was the sampling strategy?** | Whenever we use subsets, we randomly sample from the original data. |
| **Who was involved in the data collection process and how were they compensated?** | The original data collection was performed by CommonCrawl; authors from this paper were involved in retrieving it and preparing it. |
| **Over what timeframe was the data collected?** | We use all CommonCrawl dumps from 2008 to January/February 2023. |
| **Were any ethical review processes conducted?** | No. |



**PREPROCESSING**



| | |
|---|---|
| **Was any preprocessing/cleaning/labeling of the data done?** | Yes, we applied extensive preprocessing and cleaning of the data. We first filter URLs to remove adult content using a blocklist and a score system (Appendix I.1), we then use `trafilatura` [46] to extract content from pages, and perform language identification with the `fastText` classifier from CCNet [27]. After this first preprocessing stage, we filter data using heuristics from MassiveWeb [12] and our own line-wise corrections (Appendix I.2). Finally, we run extensive deduplication, removing URLs revisited across dumps (Section 3.3) and performing subsequently fuzzy and exact substring deduplication, with each stage drawing from [30]. See Section 3 for further details and Table 2 for an outline. |
| **Was the "raw" data saved in addition to the preprocessed/cleaned/labeled data?** | During development, we saved intermediary outputs from our pipeline for investigations and for ablations–intermediary outputs exist for about 5% of RefinedWeb. We did not keep intermediary outputs for the final production version of the dataset due to storage and resource constraints. |
| **Is the software that was used to preprocess/clean/label the data available?** | No. |



**USES**



| | |
|---|---|
| **Has the dataset been used for any tasks already?** | Yes, this data has been used to develop large language models: both for scientific experiments (e.g., this paper) and production use. See Almazrouei et al. [82] for details. |
| **Is there a repository that links to any or all papers or systems that use the dataset?** | On a voluntary/self-reporting basis, the HuggingFace Hub where this dataset is hosted will point to models trained using this dataset. |
| **What (other) tasks could the dataset be used for?** | RefinedWeb was built as a large-scale corpora representative of the web, and as such may see many downstream uses which are difficult to predict. |
| **Is there anything about the composition of the dataset or the way it was collected and preprocessed/cleaned/labeled that might impact future uses?** | For the public extract of RefinedWeb, we chose to only draw from the English version of the dataset, preventing multilingual applications. |
| **Are there tasks for which the dataset should not be used?** | Any tasks which may considered irresponsible or harmful. |



**DISTRIBUTION**



| | |
|---|---|
| **Will the dataset be distributed to third parties outside of the entity on behalf of which the dataset was created?** | Yes, we make a 600B tokens extract publicly available for NLP practitioners. We currently don't plan to share the full version of the dataset. |
| **How will the dataset will be distributed?** | The dataset will be made available through the HuggingFace Hub, in the `datasets` format [83]. |
| **When will the dataset be distributed?** | The dataset is available immediately. |
| **Will the dataset be distributed under a copyright or other intellectual property license, and/or under applicable terms of use?** | The public extract is made available under an ODC-By 1.0 license; users should also abide to the CommonCrawl ToU: https://commoncrawl.org/terms-of-use/ |
| **Have any third parties imposed IP-based or other restrictions on the data associated with the instances?** | Not to our knowledge. |
| **Do any export controls or other regulatory restrictions apply to the dataset?** | Not to our knowledge. |



**MAINTENANCE**



| | |
|---|---|
| **Who will be supporting/hosting/maintaining the dataset?** | The dataset will be hosted on the HuggingFace Hub, and we will release further versions if necessary based on opt-out requests. |
| **How can the owner/curator/manager of the dataset be contacted?** | falconllm@tii.ae |
| **Is there an erratum?** | No. |
| **Will the dataset be updated?** | Yes, for opt-out requests. |
| **If others want to extend/augment/build on/contribute to the dataset, is there a mechanism for them to do so?** | The license allows for the community to fork, build upon, and modify this dataset, as long as proper attribution is given. |

Table 6: **Datasheet for RefinedWeb**, following the framework introduced by [85].

# D  Falcon-RW Model Cards

| MODEL DETAILS | |
|---|---|
| **Organization** | The models were created by the Technology Innovation Institute. |
| **Model date** | Falcon-RW models were trained in December 2022/January 2023. |
| **Model type and information about training** | Falcon-RW are autoregressive Transformer models trained with a causal language modeling objective. Architecture based on GPT-3 [2], with ALiBi positional encodings [72] and FlashAttention [73]. See Section 4.1 for details. |
| **Licence** | Apache 2.0. |
| **Point of contact** | falconllm@tii.ae |

| INTENDED USE | |
|---|---|
| **Primary intended uses** | Research on large language models, and the influence of adequately filtered and deduplicated web data on the properties of large language models (fairness, safety, limitations, etc.). |
| **Primary intended users** | NLP researchers. |
| **Out-of-scope use cases** | Production use without adequate assessment of risks and mitigation; any use cases which may be considered irresponsible or harmful. |

| FACTORS | |
|---|---|
| **Relevant factors** | Falcon-RW models are trained on English data only, and will not generalize appropriately to other languages. Furthermore, as they are trained on a large-scale corpora representative of the web, they will carry the stereotypes and biases commonly encountered online. |
| **Evaluation factors** | We evaluated the toxicity of the underlying pretraining dataset and found it to be in line with common curated pretraining datasets such as The Pile (see Figure 5). Note that this only accounts for toxicity under the definition of Perspective API: "content that is rude or disrespectful". Notably, this fails to include concerns about social biases or harmfulness. |

| METRICS | |
|---|---|
| **Model performance measures** | We focus our evaluation on measuring the zero-shot generalization capabilities of our models across a wide range of tasks, leveraging the Eleuther AI language model evaluation harness [49]. |
| **Variation approaches** | Due to the costs associated with training Falcon-RW we cannot train the models multiple times and measure variability across training runs. |

| EVALUATION DATA | |
|---|---|
| **Datasets** | We evaluate zero-shot accuracy on 18 varied tasks, detailed in Table 3. |
| **Motivation** | We selected and aggregated tasks to build comparisons with other models in the literature (see Section 4.1; Appendix H.1 for details). |
| **Preprocessing** | We use the default prompts and setup of [49]. |

| TRAINING DATA | |
|---|---|
| See the dedicated datasheet in Table 6. | |

Table 7: **Model card for Falcon-RW**, following the framework introduced by [86].

## E Dataset analysis

The large-scale and diverse nature of web corpora make them difficult to document and analyse extensively; we provide some key metrics in the section, focusing on document lengths in Figure 4(a), a breakdown of the top domain names in Figure 4(b), and the distribution of toxic content in Figure 5.

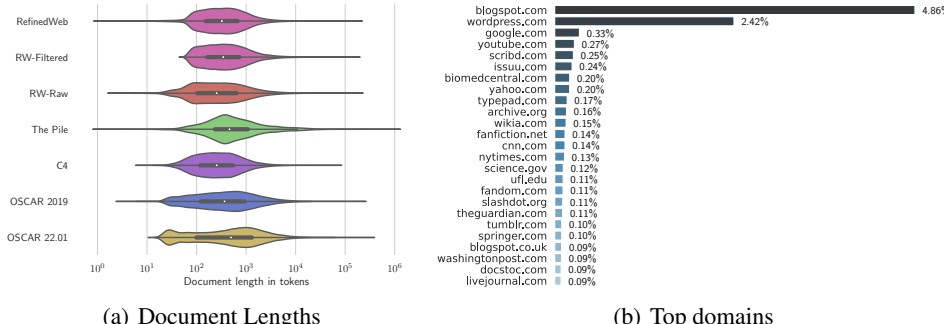

(a) Document Lengths                     (b) Top domains

Figure 4: **Make-up of RefinedWeb in document lengths (left) and top domains (right).** (a) We find the OSCAR datasets and RW-Raw to have similar document length distributions; following filtering, most of the short documents are discarded from RW-Filtered. As deduplication removes spans, it shortens documents in RefinedWeb. We note the make-up of C4 and RefinedWeb to be similar, with a longer tail of short documents for RefinedWeb. Finally, The Pile exhibit a unique make-up, with a long tail of both long (books, etc.) and short documents. (b) Top domains in RefinedWeb span from popular content platforms (Blogspot, WordPress, Tumblr, etc.), to news websites (CNN, New York Times, etc.), and include also technical content such as BioMed Central or Springer.

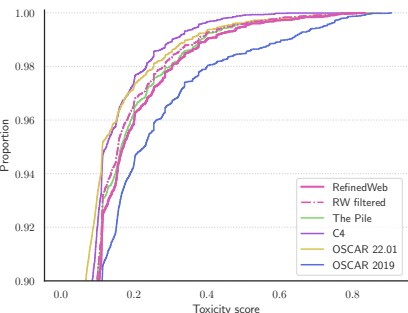

Figure 5: **Toxic content in RefinedWeb is distributed similarly to The Pile.** Cumulative proportion of documents below a given toxicity score, as evaluated by the Pespective API.

## F Multilingual RefinedWeb

**Multilingual data.** Using the language identification filter, we classify processed CommonCrawl data into 176 languages. Figure 6 shows the top 20 languages present in the data *excluding English*, based on their relative contribution in descending order. 58.20% of all documents in CommonCrawl were identified as English. We find the distribution of languages in CommonCrawl to only be partially aligned with the worldwide distribution of language speakers [87]: Russian is over-represented (2nd in CC but only 8th worldwide), Mandarin Chinese is under-represented (6-7th in CC but 2nd worldwide), and Hindi does not show-up in the top 20 despite being the 3rd most spoken.

**Processing multilingual data.** The MDR pipeline can be used to process all languages: features such as text extraction are language-agnostic, whereas specific filters such as line-wise corrections need to typically be tuned for each individual language. We also found tuning deduplication parameters for individual languages to be beneficial.

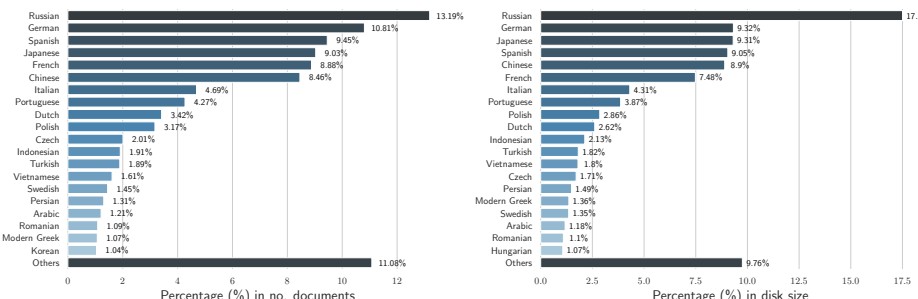

Figure 6: **The representation of languages in CommonCrawl does not align with the worldwide distribution of language speakers**. Top 20 languages (excluding English, which accounts for 58.20%) from processed CommonCrawl based on number of documents and disk size.

## G  Additional results

In this section, we present additional results obtained during the development of the Macrodata Refinement pipeline. For Appendix G.1 and Appendix G.3, these were obtained using earlier development versions of the dataset, so results are not directly comparable with the main text. For Appendix G.2, this is based on the Falcon-RW models.

### G.1  Small-scale ablations on deduplication approaches

We present results in Table 8–setup is similar to our earlier ablations, training 1B models for 30GT:

- **MinHash alone is insufficient**, it does not match the performance of exact deduplication. Conversely, combining it with exact deduplication doesn't improve performance further.
- **Masking spanned duplicates degrades performance**, systematically underperforming other approaches. Dropping and cutting spans perform similarly, although it's likely that dropping documents slightly outperforms cutting.

We chose to apply MinHash before exact deduplication, as it is easier to scale: approximate duplication acts as a pruning phase, enabling us to scale deduplication further. Finally, we choose the common option of cutting spans, as dropping resulted in even more stringent rejection rates which would have compromised our ability to collect 5 trillion tokens.

Table 8: **MinHash alone is insufficient to match the performance of exact substring deduplication, and combining the two does not significantly improve performance. Of all of the exact substring approaches, masking duplicated spans underperform, but all others exhibit similar performance.** ✓ Minhash + Exact substring-Cut corresponds to our final deduplication setup. Perplexity in bits-per-bytes on The Pile (`pile-bpb`, lower is better), zero-shot performance aggregated over LAMBADA, PIQA, and HellaSwag (`agg-dev`). Best results in **bold**, best results with minhash in underline.

| Minhash | Exact substring | pile-bpb ↓ | agg-dev-1 ↑ |
|---------|-----------------|------------|-------------|
| | RefinedWeb-Filtered | 1.11 | 43.51 |
| | Mask | 1.08 | 45.84 |
| ✓ | Mask | 1.07 | 46.28 |
| ✓ | | 1.07 | 46.57 |
| ✓ | Cut | **1.05** | 47.11 |
| | Cut | 1.06 | 47.24 |
| ✓ | Drop partial | **1.05** | 47.25 |
| | Drop any | 1.07 | 47.77 |
| ✓ | Drop any | 1.07 | 47.86 |
| | Drop partial | 1.06 | **47.97** |
| | Pile | 0.88 | 43.70 |

## G.2  Language modeling evaluation

Along with our aggregates, we also evaluated perplexity on Wikitext (Table 9). We found that models trained on RefinedWeb achieve performance close to that of models trained on The Pile. Importantly, we note that RefinedWeb does not contain any content from Wikipedia – it is explicitly filtered out at the URL level. We believe this accounts for most of the difference in perplexity, as RW models may not be familiar with the idiosyncrasies of Wikitext (e.g., layout of an article, etc.)

Table 9: **Models trained on RefinedWeb achieve performance close to models trained on The Pile on Wikitext, despite not having seen any content from Wikipedia.** Perplexity in bits-per-bytes on Wikitext (`wiki-bpb`, lower is better.)

| Model size | 1B | | 7B |
| Dataset | The Pile | RW | RW |
|---|---|---|---|
| `wiki-bpb`↓ | 0.64 | 0.66 | 0.60 |

## G.3  Does deduplication help with multiple epochs?

Earlier in this work, we outlined that to scale pretraining data, practitioners had two choices: (1) improve data collection, which is the avenue we chose to pursue; (2) train models on multiple epochs of the same data. Due to current uncertainties in the ability of larger models to sustain multiple epochs without adverse effects [41], we focused on (1). A fairly rational question regarding (2) is whether deduplication may improve the situation, and whether deduplicated data may be able to sustain more epochs without compromising model quality.

We train 1B parameters models on 30GT of RW and RW-Filtered. We keep the number of pretraining tokens fixed, but train for 1, 5, 25, and 100 epochs. This is a small-scale, limited set-up, which would have to be improved to obtain definitive results. We plot the degradation in performance compared to a single epoch in Figure 7(a) and the gap between RW and RW-F in Figure 7(b). We find that the absolute degradation is less important for RefinedWeb than for RefinedWeb-Filtered; furthermore, the gap widens with increasing number of epochs. However, we observe significant variability across tasks.

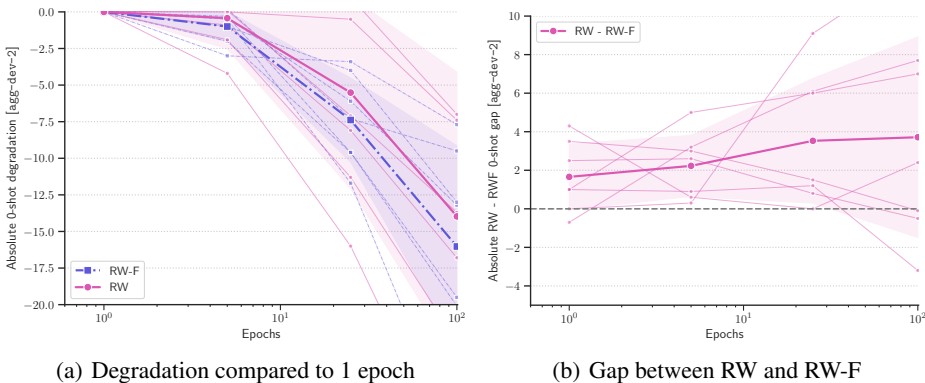

(a) Degradation compared to 1 epoch      (b) Gap between RW and RW-F

Figure 7: **Deduplication may reduce the degradation in performance incurred by multiple epochs.** However, our experiments were only performed at small-scale (1B models trained on 30GT), and we see high variability in outcomes across tasks. Zero-shot performance measured on the `agg-dev-2` aggregate (HellaSwag, PIQA, ARC, BoolQ, COPA, MRPC, SciQ). Individual curves for per-task results and 1-$\sigma$ standard deviation across all tasks in the aggregate in transparent.

Table 10: **We source evaluation results from a variety of papers across the literature, maximizing task coverage.** Although most results come from the EAI Evaluation Harness [49], results from PaLM and GPT-3 are sourced from their respective papers. Note in Figure 1 that the results from the GPT-3 paper are still ahead of results obtained through the API with the EAI evaluation harness.

| Models | Aggregates reported | Source of results | EAI eval harness? |
|---|---|---|---|
| Ours | main, core, ext | This paper | ✓ |
| BS-A&S* | main, core | [11] | ✓ |
| GPT-Neo* | main, core | [11] | ✓ |
| PaLM† | main | [22] | |
| GPT-3 API* | main, core | [11] | ✓ |
| GPT-3† | main | [2] | |
| Aleph Alpha* | core | [71] | ✓ |
| Cerebras-GPT* | core | [48] | ✓ |
| FairSeq* | core | [70] | ✓ |
| Pythia(-Dedup)* | core | [48] | ✓ |
| OPT* | core | [48] | ✓ |
| GPT-J* | core | [70] | ✓ |
| GPT-NeoX 20B* | core | [70] | ✓ |

# H  Tasks, models, and datasets from the state-of-the-art

## H.1  Task aggregates

We average zero-shot performance over diverse task aggregates, outlined in Table 3:

- `small`: small-scale ablation studies, tasks with non-zero performance for 1B parameters models trained on 30GT;
- `core`: comparisons with a wide range of models, based on the tasks reported in [48];
- `main`: tasks available in the GPT-3 and PaLM papers [2, 22];
- `ext`: tasks available in the work of the BigScience Architecture and Scaling group [11].

Detailed evaluation results are available in this dedicated spreadsheet: `https://docs.google.com/spreadsheets/d/1uOHqZVtNxe2bYmF_1lQneROFH-s6TOnjiREObV1LtEA/`. When comparing with other models, we source results from papers detailed in Table 10.

## H.2  Models

We compare against 10 series of models trained on a variety of curated corpora, presented in Table 11.

**Cerebras-GPT with $\mu$-parametrization.** The Cerebras-GPT series [48] also comes in a smaller series, up to 2.7B parameters, using $\mu$-parametrization [88]. As we found the performance of this smaller series to be close to the main series of models (see Figure 8), and as it does not include models of a similar compute scale as the ones we compare to, we do not report it in our main figures.

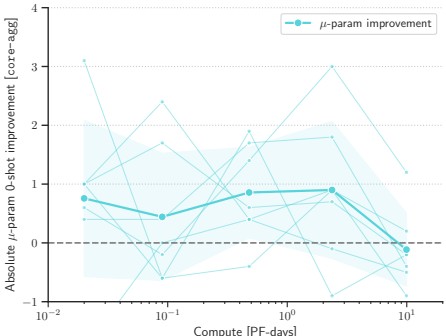

Figure 8: **$\mu$-parametrization [88] slightly improves performance in the Cerebras-GPT series [48].** Zero-shot performance on our `core` aggregate, gap between Cerebras-GPT with $\mu$-param and without. Individual curves for per-task results and 1-$\sigma$ standard deviation across all tasks in shade.

**Pythia and deduplication.** The Pythia series of models is available in two flavours: one trained on the vanilla version of The Pile, and another trained on a version deduplicated with MinHash. Performance between these two flavours was noted to minimally differ [42]; in Figure 9, we find the deduplicated version may be slightly ahead of the non-deduplicated one under our aggregate. The higher end of this improvement is broadly in line with our findings in Table 5. Nevertheless, a difference in our findings and theirs remain. We posit a few possible hypotheses:

- **Differences between curated and web data.** It is possible that web data is more sensitive to duplicates. For instance, the most common duplicates in web data (e.g., spam) may be more detrimental than the most common duplicates in curated data. This suggests a qualitative component to deduplication that we have not studied in this work.
- **Differences in deduplication pipeline.** Because [42] uses the MinHash settings from [30], they are mostly identical to ours. However, we also apply exact deduplication: while their deduplication incurs a 30% reduction in size, our deduplication is more aggressive, resulting in a 45% reduction in size. This may explain why our results in Table 5 show a stronger gain from deduplication than theirs in Figure 9.
- **Differences in pretraining.** Finally, we note that [42] chooses to perform a partial extra epoch on the deduplicated data to reach 300GT, while we always perform a single epoch. Their setting corresponds to a data-constrained scenario, which is more realistic for the curated data they study; for us, web data is plentiful, so deduplication never truly limits the size of the datasets we can use.

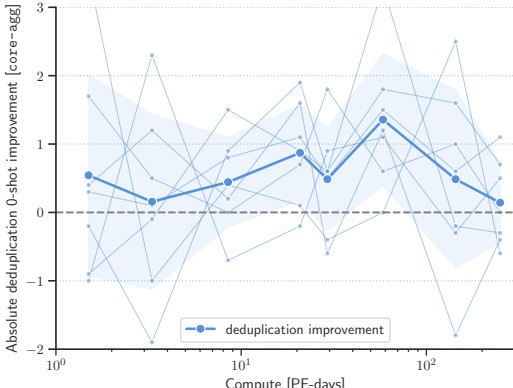

Figure 9: **In our `core` aggregate, deduplication brings a small improvement to the Pythia suite [42].** Zero-shot performance on our `core` aggregate, gap between Pythia trained on the deduplicated and vanilla Pile. Individual curves for per-task results and 1-$\sigma$ standard deviation across all tasks in the aggregate in transparent.

### H.3    Datasets

We extend on Table 1 in Table 12, providing details on the filtering and deduplication strategies used across the litterature.

Table 11: **Full-scale models trained on RefinedWeb (Falcon-RW) and other models from the state-of-the-art.** Across models trained on The Pile, the Pythia models are the closest to our achitecture: they use FlashAttention with rotary embeddings–with for only notably exception the use of parallel attention and feedforward for their models. Training budget $C$ in PF-days calculated using $C = 6ND$, with $N$ the number of parameters, and $D$ the pretraining dataset size [5].

| Series | GPT-3 (paper)[†] | | GPT-3 (API)* | | BigScience* | PaLM[†] | Ours | | |
|---|---|---|---|---|---|---|---|---|---|
| **Model** | XL | XXL | babbage | curie | BS-A&S | PaLM-8B | Ours (Pile) | Ours (RW) | |
| **Dataset** | GPT-3 | GPT-3 | GPT-3 | GPT-3 | Pile | PaLM | Pile | RW | RW |
| **Params.** | 1.3B | 6.7B | 1.3B | 6.7B | 1.3B | 8.6B | 1.3B | 1.3B | 7.5B |
| **Pretraining** | 300GT | 300GT | 300GT | 300GT | 300GT | 780GT | 350GT | 350GT | 350GT |
| **PF-days** | 27 | 140 | 27 | 140 | 27 | 466 | 32 | 32 | 182 |
| **Citation** | | | [2] | | [11] | [22] | This paper | | |

| Series | EleutherAI* | | | Pythia* |
|---|---|---|---|---|
| **Model** | GPT-Neo | GPT-J | GPT-NeoX 20B | Pythia(-Dedup) |
| **Dataset** | Pile | Pile | Pile | Pile (dedup) |
| **Params.** | 1.3B | 6.7B | 20B | 70M-12B |
| **Pretraining** | 380GT | 402GT | 472GT | 300GT |
| **PF-days** | 34 | 187 | 656 | 1.5 - 250 |
| **Citation** | [77] | [74] | [70] | [42] |

| Series | Aleph Alpha* | Cerebras-GPT* | OPT* | FairSeq* |
|---|---|---|---|---|
| **Model** | Luminous | Cerebras-GPT | OPT | FairSeq |
| **Dataset** | *undisclosed* | Pile | Pile (subset) + curated | curated |
| **Params.** | 13B | 111M-13B | 125M - 175B | 1.3 - 13B |
| **Pretraining** | 400GT | 2 - 257GT | 300GT | 300GT |
| **PF-days** | 361 | 0.02 - 232 | 3 - 3646 | 27 - 271 |
| **Citation** | [71] | [48] | [43] | [76] |

Table 12: **Common massive web-scrape and LLM English datasets.** Datasets such as OSCAR and C4 also have significant multilingual versions, which have enjoyed wide adoption [89]. For OSCAR, the size corresponds to the non-deduplicated version, and is estimated from the number of words x0,75 (average number of words per tokens).

| General information | | | | Web data | | | | | |
|---|---|---|---|---|---|---|---|---|---|
| Dataset | Notable models | Size | Availability | Web | HTML extraction | Language ID | Heuristics | Content filtering | Deduplication |
| **MASSIVE WEB DATASETS** | | | | | | | | | |
| C4 [9] | T5 [9] | ~360GT | Public | 100% | .WET files | Document-level w/ langdetect | Document and line-level | Rules-based: code, NSFW | **Exact:** three sentences span |
| OSCAR 21.09 [8] | | ~370GT | Public | 100% | .WET files | Line-level w/ fastText [28] | Line < 100 characters | None | (optional) **Exact:** per line (~55% removed) |
| OSCAR 22.01 [90] | | ~283GT | Public | 100% | .WET files | Document-level w/ fastText [28] | Line-level, optional document-level | Optional NSFW blocklist | (optional) **Exact:** per line |
| **CURATED DATASETS** | | | | | | | | | |
| GPT-3 [2] | | 300GT | Private | 60% | Unknown | Unknown | Unknown | fastText trained on HQ-data | **Fuzzy:** min-hash with 10 hashes (~10% removed) |
| The Pile [10] | GPT-J [74], GPT-NeoX-20B [70], Pythia [42], Cerebras-GPT [48] | ~340GT | Public | 18% | jusText [91] | Document-level w/ pycld2 [92] | None | fastText on curated crawl | **Fuzzy:** min-hash with 10 hashes, sim. treshold 0.5 (~26% removed) |
| MassiveWeb [12] | Gopher [12], Chinchilla [4] | 1,400GT | Private | 48% | Custom | Unknown | Document-level | SafeSearch | **Exact & fuzzy:** exact documents, minhash w/ sim. treshold 0.8 |
| PaLM [22] | | 780GT | Private | 27% | Unknown | Unknown | Document-level | ML-based filter on HQ data | Unknown |
| **OURS** | | | | | | | | | |
| **REFINEDWEB** | Falcon-RW | 5,000GT | 500GT Public | 100% | trafilatura [46] | From CCNet [27] | Document and line-level | URL blocklist | **Exact & fuzzy** |

# I Details of the Macrodata Refinement pipeline

## I.1 URL filtering

As discussed in Section 3.1, we base our filtering of adult documents only on the URL itself, and not on the content of the documents. This design choice was motivated by: (1) challenges in avoiding overfiltering content from minorities when using ML-based classifiers on the content of documents [44]; (2) NSFW words block-list applied on content (such as the one used in C4) also resulting in overfiltering of legal and medical content [31].

Our URL filtering focuses on finding domains that are related to adult content, that may be harmful to users, or that are very likely to contain mostly unstructured text/spam (e.g., file hosting websites). First, we aggregated a list of 4.6M domains, detailed in Appendix I.1.1, that we explicitly ban; then, we built a simple URL scoring system, based on matching subwords in the URL against a list of words we curated (see Appendix I.1.2). We curated this list of words based on manual inspection, cross-referencing results with pages surfaced by ToxicBERT as being outliers in toxicity [93].

### I.1.1 URL Blocklist

**Origin of the list.** We use an aggregated list[1] of about 4.6M URLs that we explicitly ban. This list is broken in categories (e.g. pornography, gambling); we outline the categories we selected in Table 13. The list is regularly updated, with an original intended usage as a blocklist for universities.

**Curation.** We noticed the list blocked a number of domains inappropriately; while these domains were few ($<$100), they accounted for a significant portion of the data filtered by the list, as these were rather prolific domains, with thousands of pages of content. To identify these false positive domains, we applied the blocklist to a subset of 832M pages. 6.04M (0.73%) pages matched with the blocklist, and the number of occurrences per URL ranged from 1 to 79k. We manually inspected all URLs matched more than 4k times, which represented an appreciable portion of the dataset. We found a number of benign domains, such as pop culture news websites, or blogging platforms, which we removed from the list.

Table 13: **We select categories likely to contain adult or malicious content, as well as spam or unstructured text.**

| Category | Description | Number of links |
|---|---|---|
| adult | adult websites: from eroticism to hard pornography | 4516478 |
| phishing | phishing websites, malwares, etc. | 42445 |
| dating | dating websites | 3829 |
| gambling | online casino | 1365 |
| filehosting | websites hosting files, videos, pictures, music | 909 |
| ddos | websites related to ddos attacks | 421 |
| agressif | hate, racism, etc | 390 |
| chat | online chat websites | 244 |
| mixed adult | websites with some adult content | 153 |
| arjel | French regulated gambling websites | 69 |

### I.1.2 URL Scoring with a Word-List

To score URLs, we used three matching patterns based on a soft, hard, and strict violation word-list:

- **Strict subword matching**: http://foobann.edsub-wo.rdbar.com/any/bar, matching words such as xvideos, groupsex;
- **Hard whole word matching**: http://www.foo.bannedword-bar.com, with words such as porn, xxx, orgy;

---
[1] https://dsi.ut-capitole.fr/blacklists/

- **Soft words matching**: http://www.foo.soft1-bar-soft2.com, with "softer" words such as sex, webcam, escort.

Each list is associated with a different level of severity: for the strictest one (strict subword matching), we ban any URL matching a banned word in its substrings (as fraudulent websites may attempt to escape similar recognition schemes by breaking-up adult keywords); for the hard whole word matching, we ban URLs with a whole word matching in the list; finally, a minimum of two matches are required with the soft word matching.

We curated the lists based on manual inspection of the data, informed by top hits reported by ToxicBERT. For the strict subword matching, we included words that were unequivocally related to adult content (e.g., groupsex). We avoided partial unclear matches (e.g., ass), that may be part of neutral words (e.g., massachusetts). In the soft word list, we included words that do not constitute a sufficient reason to discard the document on their own, but which are suspicious when multiple words from the list result in a match. This helped with keeping medical or legal content unaffected (e.g., a single match of dick).

### I.1.3 Excluded High Quality Sources

Since our paper focuses on the study of RefinedWeb alone, we chose to exclude common online sources of curated data from it. This serves two objectives: (1) it strengthens our results, by ensuring that RefinedWeb doesn't end-up actually being made mostly of known high-quality sources (e.g., Wikipedia represents a significant portion of C4); (2) future works may be interested in combining RefinedWeb with existing curated copora, which would require further deduplication if they are included in RefinedWeb. Accordingly, we remove common sources used in The Pile [10] from RefinedWeb. The full list of curated data sources domains that we blocked is in Table 14.

Table 14: **RefinedWeb is stripped from common so-called high-quality sources to simplify combining it with existing curated corpora**. This blocklist is applied at the URL filtering stage, along with the adult content blocklist.

| Curated data source | Domain name blocked |
| --- | --- |
| arxiv | arxiv.org |
| AskUbuntu | askubuntu.com |
| StackOverflow | stackoverflow.com |
| | stackapps.com |
| | stackexchange.com |
| | mathoverflow.net |
| NIH Abstracts | exporter.nih.gov |
| | ncbi.nlm.nih.gov |
| Github | github.com |
| Ubuntu IRC | irclogs.ubuntu.com |
| HackerNews | news.ycombinator.com |
| FreeLaw | courtlistener.com |
| Reddit | reddit.com |
| Europarl | statmt.org |
| United States Patents | uspto.gov |
| Wikipedia | wikipedia.org |

### I.2 Line-wise filtering

Despite the improvements brought forth by running text extraction with Trafilatura, we found that a number of irrelevant lines still seeped through. These lines are usually related to navigation menus, call to actions, or social media counters. Following manual inspection of the data, we devised a line-wise filtering strategy. We analyse documents line-by-line, and discard or edit the lines based on the following rules:

- If it is mainly composed of uppercase characters (discard);
- If it is only composed of numerical characters (discard);
- If it is a counter (e.g. `3 likes`) (discard);
- If it only contains one word (discard);
- If it is short ($\leq 10$ words) and matches a pattern (edit):
  - At the beginning of the line (e.g. `sign-in`);
  - At the end of the line (e.g. `Read more...`);
  - Anywhere in the line (e.g. `items in cart`).

Finally, if the words in the flagged lines represent more than $5\%$ of the total document words, the document is discarded. We derived these filters through manual inspection of the data, and note that they require adaptation across languages.

## I.3 Deduplication

We make use of the two deduplication methods described in [30]: EXACTSUBSTR and NEARDEDUP (detailed in Appendix I.3.1 and Appendix I.3.2; see Appendix J for samples of duplicates).

We start with the most scalable approach, NEARDEDUP. We remove similar documents by applying MinHash [34], whereby a signature/sketch supporting efficient approximate similarity queries is computed for each document in the dataset, and document pairs with a high *n*-gram overlap are identified.

We then use EXACTSUBSTR, leveraging the implementation from [30][2], to identify ranges of exact duplicate text of at least 50 tokens. We experiment with three different approaches for these ranges: EXACTSUBSTR-CUT, where we remove them from the original text, as done in the original implementation; EXACTSUBSTR-MASK, where the dataset is unchanged but we do not compute the loss on the duplicated ranges; and EXACTSUBSTR-DROP, where we simply drop an entire document if the duplicated ranges make up more than a certain percentage of its content.

We present small-scale ablations around these different approaches in Appendix G.1.

### I.3.1 MinHash Approximate Matching

We employ MinHash to find approximate duplicate documents in our web corpora at a very large scale. This technique allows us to identify templated pages or otherwise very similar content where most of the interspersed duplicated sections are small enough to not be identified by exact matching methods (anything smaller than 50 tokens).

**Signing.** We start by normalizing the content to increase recall: punctuation is removed, text is lowercased, NFD Unicode normalization is applied, accents are removed, and all whitespace is normalized. We tokenize the resulting text using the GPT-2 tokenizer [17] and obtain the set of unique *n*-grams for each document. Hash functions are used to obtain a signature for each document: for each hash function, the smallest value is kept from hashing every unique *n*-gram in the document. If two documents are similar, then there is a high probability that they will have the same minimum hash (MinHash) for at least some of the hash functions used [34]. The ratio of matching hashes between two documents approximates the Jaccard Similarity [94] of the sets of their unique *n*-grams (the sets being $d_i$ and $d_j$):

$$J(d_i, d_j) = \frac{|d_i \cap d_j|}{|d_i \cup d_j|} \tag{1}$$

**Matching.** Since comparing MinHash signatures between every possible document pair is computationally expensive, we apply a locality sensitive hashing version of MinHash, MinHash LSH. A document signature is split into *r* buckets, each with *b* minhashes. Documents are indexed by these *b*

---

[2]https://github.com/google-research/deduplicate-text-datasets

minhashes on each of the $r$ buckets, and we mark two documents as duplicates if their $b$ minhashes are exactly the same on at least one of the buckets. These two parameters, $b$ and $r$, will determine the probability that similar documents will be detected. For two documents $i$ and $j$ whose ratio of matching hashes between their MinHash signatures is $s_{i,j}$, the probability that there is a match in a given bucket is $s_{i,j}^b$; the probability that there isn't a match in any of the buckets is $(1 - s_{i,j}^b)^r$; and finally that there is a match in at least one of the buckets:

$$P = 1 - (1 - s_{i,j}^b)^r \tag{2}$$

We use the same parameters as [30]: $n = 5$ (5-grams); $b = 20$ and $r = 450$. This means that for each document, we compute a total of 9000 minhashes, and that the probability that a document pair with similarity 0.75 or 0.8 will be marked as duplicates will be $76\%$ and $99.4\%$ (respectively), diminishing rapidly for smaller similarity values.

Finally, we cluster documents across all buckets — if documents A and B match in one bucket and B and C in another, A-B-C becomes a cluster. We randomly remove all but one of the documents in each cluster.

[30] also proposed filtering down on false positives by computing the real Jaccard similarity, or other metrics such as the edit similarity between identified document pairs. Given the large amount of data we have available across all of CommonCrawl, and that our main concern is improving recall, we decided to skip this additional step.

### I.3.2 Exact substring deduplication

We make use of the EXACTSUBSTR implementation publicly released by [30] for exact text matching. We apply exact substring deduplication to data that has already been deduplicated by MinHash, reducing by nearly 40% size of the dataset on which we have to operate. EXACTSUBSTR will find long strings of text that are present, character for character, across multiple documents. Some of these may have escaped the earlier stage of approximate deduplication: they might not constitute a big enough portion of the document; one document might have repeated sections sourced across many different documents; or they may simply not have been found due to the approximate nature of MinHash.

**Finding duplicates.** EXACTSUBSTR concatenates all the documents in the dataset to create a single long text sequence; then, it builds a suffix array [33] in linear time—an array of the indexes to a lexicographical ordering of all the suffixes in the sequence. Finally, duplicate sequences can also be found in linear time using the suffix array, by simply traversing the ordered list of suffixes and comparing the beginning of each pair of two consecutive suffixes.

We apply the same normalization and tokenization as for MinHash to the content of our documents before concatenating them. One important difference is that reversibility is important: for MinHash, we were discarding entire documents, and thus never relying on the normalized+tokenized representation for downstream use. Here, once we have identified duplicate normalized+tokenized spans, we need to revert to the original span to remove it. Accordingly, we include normalization in the tokenization process, and validate that the process is reversible.

If a match is longer than 50 tokens, there will be multiple overlapping duplicated ranges. These overlapping duplicated ranges in the concatenated dataset sequence are merged before we save them to a file. We then take these ranges and retrieve the original document that produced them, obtaining the character substrings corresponding to the duplicated token ranges.

**Removing duplicates.** We considered applying the following transformations to the duplicate spans:

- EXACTSUBSTR-CUT: we remove the duplicated spans, and discard documents where there are fewer than 20 non-duplicated characters left–this is the vanilla setting used by [30];

- EXACTSUBSTR-MASK: we loss-mask the duplicated spans, preventing a loss from being computed on the duplicated text during pretraining, and discard documents where there are fewer than 20 non-masked characters left.
- EXACTSUBSTR-DROPPARTIAL: if more than 20% of the document is duplicated, we remove the entire document;
- EXACTSUBSTR-DROPANY: we drop any document with a duplicated span in it.

Broadly speaking, EXACTSUBSTR-CUT might remove text mid-sentence resulting in disconnected text; EXACTSUBSTR-MASK does not have this issue, but might be less efficient as a significant portion of the training tokens will not directly contribute to updating the model's weights; EXACTSUBSTR-DROP might still keep considerable duplicated sections in its PARTIAL version, especially on larger documents, while the ANY version might be overly aggressive. Following ablations in Appendix G.1, we choose to stick with the vanilla approach, EXACTSUBSTR-CUT.

Note that in all cases, while MinHash keeps one copy of the duplicated documents, our exact deduplication removes all copies of the duplicated span.

### I.4 Execution environment

Most data processing took place in large CPU clusters, with 100-250 AWS c5.18xlarge instances; each instance has 72 vCPUs and 144 GiB of memory. We usually run with 10,000-20,000 vCPUs in the cluster, enabling rapid parallel processing.

For EXACTSUBSTR, the entire dataset being deduplicated needs to be loaded onto memory: we leveraged the AWS x2iedn instances, which come with up to 2 TiB of memory in a single instance.

## J  Deduplication samples from RefinedWeb

### J.1  MinHash clusters

We report the 8 largest duplicate clusters found by MinHash in Table 15 – each spanning hundreds of thousands of documents. We also found a large number of duplicate document pairs to be due to different URL GET parameters not resulting in significantly different content. An example of this behaviour can be seen in the URLs presented in Table 16.

Table 15: **Top-8 largest MinHash clusters found when building RefinedWeb.** We cut some of the longest samples in the interest of readability, only keeping a brief description.

| Description | Example document |
|---|---|
| Wordpress sitemap notice generated by the Google Sitemap Generator Plugin | This is a XML Sitemap which is supposed to be processed by search engines which follow the XML Sitemap standard like Ask.com, Bing, Google and Yahoo. It was generated using the WordPress content management system and the Google Sitemap Generator Plugin by Arne Brachhold. You can find more information about XML sitemaps on sitemaps.org and Google's list of sitemap programs. This file contains links to sub-sitemaps, follow them to see the actual sitemap content. |
| Cloudflare notice to enable Javascript | |
| Templated disability notice, with different phone numbers across pages | Welcome to our website! As we have the ability to list over one million items on our website (our selection changes all of the time), it is not feasible for a company our size to record and playback the descriptions on every item on our website. However, if you are an American with a disability we are here to help you. Please call our disability services phone line at [redacted] or [redacted] during regular business hours and one of our kind and friendly personal shoppers will help you navigate through our website, help conduct advanced searches, help you choose the item you are looking for with the specifications you are seeking, read you the specifications of any item and consult with you about the products themselves. There is no charge for the help of this personal shopper for any American with a disability. Finally, your personal shopper will explain our Privacy Policy and Terms of Service, and help you place an order if you so desire. |
| Templated cookies notice | |
| Templated domain name for sale page | |
| `www.metoperashop.org` and sub-URLs, with content changes but always the same (large) footer | |
| Different pages across more than 80 different domain names but with a common section | DC Customers also liked: Special event items are produced by manufacturers only after the outcome of a game or event. These are advanced sale items and will ship immediately after they are received in our warehouse. Manufacturer direct items are shipped directly from the manufacturer. These items are not available for international or expedited shipping. Customized items can be personalized with options such as your name, your favorite number, and/or designs. Some options may be limited by league rules. |
| `http://www.boxofficemojo.com/daily` and sub-URLs | |

Table 16: **URL with different GET parameters don't always result in significantly different page content.**

| | |
|---|---|
| `http://gamesandbiz.blogspot.com/2010/` `07/bad-reviews-can-hurt-game-sales.ht` `ml?showComment=1278486430242` | `http://gamesandbiz.blogspot.com/2010/` `07/bad-reviews-can-hurt-game-sales.ht` `ml?showComment=1278499674195` |
| `https://www.ocean-oxygen.org/home;jse` `ssionid=1E3290E84F668552FAC643D0A8F81` `BEC?p_p_id=122_INSTANCE_Zy6zjkRLAg7v&` `p_p_lifecycle=0&p_p_state=normal&p_p_` `mode=view&p_p_col_id=column-2&p_p_col` `_pos=1&p_p_col_count=6&p_r_p_56423352` `4_resetCur=true&p_r_p_564233524_categ` `oryId=1346016` | `https://www.ocean-oxygen.org/home?p_p` `_id=122_INSTANCE_Zy6zjkRLAg7v&p_p_lif` `ecycle=0&p_p_state=normal&p_p_mode=vi` `ew&p_p_col_id=column-2&p_p_col_pos=1&` `p_p_col_count=6&p_r_p_564233524_reset` `Cur=true&p_r_p_564233524_categoryId=1` `346016` |

## J.2 Exact substring matches

Examples of exact matches found by exact substring deduplication can be seen in Table 17.

Table 17: **Matches found by exact substring deduplication** (in *italics*).

| | |
|---|---|
| it appears there is a transfer of ranking signals in this relationship. Supporting this finding is a quote from Google's guidelines: *Using JavaScript to redirect users can be a legitimate practice. For example, if you redirect users to an internal page once they're logged in, you can use JavaScript to do so. When examining JavaScript or other redirect methods to ensure your site adheres to our guidelines, consider the intent. Keep in mind that 301 redirects are best when moving your site, but you could use a JavaScript redirect for this purpose if you don't have access to your website's server.* NOTE: Their experiment is based on a live page with status code 200 and NOT an inactive page. So if you want to implement this for legacy | Some examples of sneaky redirects include: - Search engines shown one type of content while users are redirected to something significantly different. - Desktop users receive a normal page, while mobile users are redirected to a completely different spam domain. *Using JavaScript to redirect users can be a legitimate practice. For example, if you redirect users to an internal page once they're logged in, you can use JavaScript to do so. When examining JavaScript or other redirect methods to ensure your site adheres to our guidelines, consider the intent. Keep in mind that 301 redirects are best when moving your site, but you could use a JavaScript redirect for this purpose if you don't have access to your website's server.* |
| Find Palm Beache FL homes for sale and other Palm Beach real estate on homesofthepalm-beaches.com. Browse and search Palm Beach houses, condos, townhomes and single-family homes by community , building, or location. *Our extensive database of real estate listings provide the most comprehensive property details including home values, features and local school and neighborhood info so you can be sure that you have nearly all the facts you need upfront. Search* home-softhepalmbeaches.com today! Want a closer look at what other Palm Beach properties are available? | Search Stuart houses, condos, townhomes and single-family homes by price and location. *Our extensive database of real estate listings provide the most comprehensive property details including home values, features and local school and neighborhood info so you can be sure that you have nearly all the facts you need upfront. Search* Stuart Listings today! Want a closer look at what other Stuart properties are available? Also search our listings for the Newest Stuart Listings and Stuart Homes with Price Reductions now. Stuart FL Homes for Sale - Stuart Real Estate Listings FREE to search Stuart Property |
| *To find the correct size you should measure your foot from the heel to the toe point. Add approximately 1 - 1,5cm to get the actual inner sole length. Measure both feet and fit shoes to the larger foot. Measure feet at the end of the day, when your feet are at their largest.* Lente shoes are women's easy slip-on leisure shoes for everyday use. These lightweight shoes have a breathable textile mesh upper made of recycled PET bottles and cool Lycra lining. | *To find the correct size you should measure your foot from the heel to the toe point. Add approximately 1 - 1,5cm to get the actual inner sole length. Measure both feet and fit shoes to the larger foot. Measure feet at the end of the day, when your feet are at their largest.* Enjoy your summer days with Masera leisure sneakers. These low-cut women's sneakers are extremely lightweight thanks to phylon midsole and breathable textile mesh upper |
| This bandana makes the perfect addition to every fur babies birthday collection! With its sparkly crown pattern, your pup will be ready for every birthday celebration! *With snaps for security, this bandana is made with love, down to the very last stitch ! Fabric: cotton Care Instructions: Hand wash only, iron as needed, on low heat Always supervise your pup while wearing Faithful Paws Co. accessories, as it could become a choking hazard if consumed.* | This bandana makes the perfect addition to every fur babies summer collection! With its vibrant watercolor popsicle pattern, your pup will be ready for every summer cookout! *With snaps for security, this bandana is made with love, down to the very last stitch ! Fabric: cotton Care Instructions: Hand wash only, iron as needed, on low heat Always supervise your pup while wearing Faithful Paws Co. accessories, as it could become a choking hazard if consumed.* |