# OpenReview forum: "The RefinedWeb Dataset for Falcon LLM: Outperforming Curated Corpora with Web Data Only"
_NeurIPS.cc/2023/Track/Datasets_and_Benchmarks — NeurIPS 2023 Datasets and Benchmarks Poster_

### Official Review · Reviewer_TYb3 · 2023-07-22
**A step forward in the open source community's understanding and utilization of LLM pre-trained web data**

**Rating:** 7
**Confidence:** 3

**Strengths:**


1. Better scalability: Using filtered and deduplicated web data for model training can significantly reduce the reliance on curated corpora, which have scalability limitations.

2. Good performance on LLM pre-training: The study shows that models trained using these extensive web-only datasets can outperform models trained on curated corpora.

3. Resource contribution: The authors have created and released the RefinedWeb dataset (500B subset). This is a valuable contribution to the machine learning community, which would benefit resource-constrained researchers and projects.

4. Toxicity: According to the perspective API's toxicity definition, the toxicity level of RefinedWeb is comparable to that of The Pile, which suggests a reasonable content quality.


**Additional Feedback:**

Plz see details above


**Clarity:**

This paper is well-written and easy to follow. The authors clearly presents their contribution and the significance of their work, providing good readability.


**Correctness:**

The paper conducts a lot of experiments and most of them are solid.


**Documentation:**

This work provides sufficient information in terms of data collection, organization, availability and maintenance, together with a URL that is publicly accessible.


**Opportunities For Improvement:**

1. Bias Analysis: The study acknowledges that their toxicity analysis does not cover issues with social biases or harmfulness, and merely assumes that the pipeline is unlikely to introduce more bias than other datasets. This leaves room for further research to deeply analyze bias within RefinedWeb.

2. Epochs Dependency: Instead of looking for "unique" tokens to create a trillion-scale pretraining dataset, one could repeat data over multiple epochs. The paper notes that further research may reveal insights about the trade-offs for using deduplication with multiple epochs and how the data's quality may factor in.

3. Deduplication Impact: The paper refers to another study that found limited performance impact from deduplicating existing datasets like The Pile, suggesting that the performance benefits may vary depending on dataset specifics. Thus, further research is encouraged in this area.


**Relation To Prior Work:**

This work provides clear explanation to how it differs from the existing contributions in their related work section


**Summary And Contributions:**

This paper addresses the scalability challenges of curating "high-quality" corpora for training large language models (LLMs), especially those that require pretraining on trillions of tokens. The authors challenge the conventional belief that a mixture of filtered web data and curated corpora is necessary for effective LLM training.

Specifically, the authors present RefinedWeb, a filtered and deduplicated dataset extracted from the web, which includes 5T tokens from CommonCrawl (CC). Despite extensive filtering, this web data source remains abundant and viable for producing competitive models.
The contributions mainly include (1) viability and effectiveness of using uncurated but adequately filtered and deduplicated web data; (2) how to create the web datasets for LLM; and (3) its publicly released 500B version.

---

> ### Author Response · Authors · 2023-08-17
> **Answer to Reviewer TYb3**
>
> We thank Reviewer `TYb3` for their positive assessment of our work and for their well-structured review.
>
> We agree with the three limitations (bias analysis, epochs dependency, and deduplication impact) raised by the reviewer, and have discussed them in our limitations section. If the reviewer has additional questions and concerns, we would be happy to answer them during the discussion period.
>
> On 2. and 3., we would like to add:
>
> * For **epochs dependency**, [1] still found a small degradation even from performing 2-4 epochs (past 4 epochs, the degradation is significant). Their study is also focused on upstream training loss, which may not fully capture zero-shot behaviour. Hence, this motivates the need to have as large of a base dataset as possible to: (1) reduce the need to rely on multiple epochs; (2) increase the maximum dataset size that can be reached with 4 epochs. Notably, we note that with RefinedWeb English and its 5 trillion tokens, that limit would be placed at ~20T tokens. While this sounds large, this is close to some of the numbers estimated for GPT-4--hence the value of pursuing work on highly scalable web crawled data.
>
> * For **deduplication impact**, we were able to improve on our side the performance on The Pile (Appendix G.)
>
>
> [1] Muennighoff et al., 2023. Scaling Data-Constrained Language Models.

---

### Official Review · Reviewer_om9q · 2023-07-23

**Rating:** 7
**Confidence:** 4
**Correctness:** Yes, as far as I can tell
**Clarity:** Yes

**Strengths:**

- The paper is well written and thoughtfully organized, including throughout the appendix.
- It is full of interesting details about the dataset and filtering process
- The evaluation is thoughtful and well-executed, and includes training several models from scratch, evaluating other models with the same codebase, and reporting results directly from other works. This gets around many of the core challenges with LM evaluations and comparisons. Additionally, it’s nice to have different evaluation splits, including the small split to get signal for smaller models.
- Even though the story with respect to the Pythia models isn’t cleanly in line with this work’s, I appreciate that it is discussed openly (and provides future questions to investigate)


**Additional Feedback:**

N/A

**Documentation:**

Yes

**Limitations:**

Yes

**Opportunities For Improvement:**

- “We publicly release an extract of 600 billion tokens” Why only a small fraction? It would be nice to say whether this is due to a resource constraint (in which case others may be able to assist) or due to commercial or safety concerns (in which case it would be better to say so openly).
- It seems like one setting where filtering would be useful is to encourage specific capabilities—e.g. additional coding data, scholarly information from textbooks or academic papers, etc. It could be worth discussing this (whether or not you agree or disagree).
- It might be nice to contextualize e.g. a 1% or 3% gain (in Table 3 or the appendix)  in terms of how much compute / data is being saved. Right now it’s a bit hard to appreciate.
- It might be nice to include results from Falcon-40B and larger models from other groups on graphs / tables where appropriate (even if just in the appendix)
- I’d like more discussion about the Oscar 21.09 results. It’s surprising that OSCAR-21.09 performs so well—just 1% worse. Additionally, it seems that filtering/deduplication provides very small benefits for the dataset (and deduplication appears not to help C4 much at all either). Do you have any insights here?


**Relation To Prior Work:**

Yes

**Summary And Contributions:**

This work describes the construction and composition of the RefinedWeb dataset and its main finding that public corpora and stringent deduplication / filtering can match performance on curated data.

---

> ### Author Response · Authors · 2023-08-17
> **Answer to Reviewer om9q: domain-specific performance (1)**
>
> We thank reviewer `om9q` for their positive assessment of our work and for their thorough review.
>
> We address below questions and concerns brought-up by the reviewer; we have broken down our answer in topic-specific messages. We will also soon be submitting a revised version of the paper reflecting these comments.
>
>
> ## Domain-specific performance
>
> > It seems like one setting where filtering would be useful is to encourage specific capabilities—e.g. additional coding data, scholarly information from textbooks or academic papers, etc
>
> We actually agree with this point, and **will add a paragraph to our limitations section to discuss our view more in-depth**. Taking from our answer to Reviewer `2Ywv`:
>
> **On simple technical/medical tasks, we found RefinedWeb to be competitive with The Pile, despite not including explicitly such data**.
>
> |                 | OpenAI-babbage | Ours-Pile | Ours-RefinedWeb |
> |-----------------|:--------------:|:---------:|:---------------:|
> | PubMedQA [acc.] |      61.2      |    60.8   |     **62.5**    |
> | SciQ [acc.]     |      86.7      |     88    |     **89.9**    |
>
> We find that despite The Pile incorporating domain-specific medical knowledge and scientific papers, it does not outperform models trained with RefinedWeb. We attribute this to two factors: (1) web data is broad, and likely still contain some medical and technical content, so our models are not entirely blind in these domains; (2) filtering and deduplication fundamentally improve the capabilities of models, and for these tasks, the models may benefit more from a generic improvement at Q/A than from domain specific knowledge.
>
> **However, our evaluation setup remains strongly focused on general NLP capabilities, and does not capture well maths/code performance.** For these datasets, it is common to add specific datasets which can be scrapped/generated in large quantities -- similarly to web data. We do not believe web data alone to be sufficient to train an adequate code model; code data is required. For instance, in Falcon-40B, we added permissively licensed code data from GitHub on top of RefinedWeb to elicit code abilities.
>
> We will add a discussion in our limitations section on domain-specific tasks (code, mathematics, etc.) Overall, the point of our paper is more to challenge the view that curated data is necessary for performant general language models (as was often brought up based on The Pile/GPT-3). We however agree that domain-specific data can be valuable for domain-specific tasks (especially when the domain is vastly different from natural language, as is code).

---

> ### Author Response · Authors · 2023-08-17
> **Answer to Reviewer om9q: contextualising zero-shot improvements (2)**
>
> ## Contextualising zero-shot improvements
>
> > It might be nice to contextualize e.g. a 1% or 3% gain (in Table 3 or the appendix) in terms of how much compute / data is being saved
>
> This is an important point also brought-up by `GCg1`, we will add a short discussion in the main text on this subject. **On aggregated zero-shot benchmarks, small (~1%) improvements are significant.** This is best illustrated in the example we gave to reviewer `GCg1`:
>
> In ablations reported in Table 3, we find that the 1.3B parameters model trained on 27B tokens of RefinedWeb outperforms a 3B parameters model trained on 60B tokens tokens on OSCAR-22.01. **Accordingly, training on better data here leads to a 4x reduction in compute costs!**
>
> Data filtering, cleaning, and deduplication improves performance in a way that could only be compensated by increasing training budgets significantly: not just by a few percents, but by 25-300% depending on the quality of the original dataset. This is because gaining a few percents in zero-shot aggregate performance is actually difficult, and translates significant improvements in the model. Compute alone cannot efficiently compensate for bad data, which motivates for the creation of high-quality datasets such as RefinedWeb.

---

> ### Author Response · Authors · 2023-08-17
> **Answer to Reviewer om9q: OSCAR/C4 performance and miscellaneous questions (3)**
>
> ## OSCAR/C4 performance
>
> > I’d like more discussion about the Oscar 21.09 results. It’s surprising that OSCAR-21.09 performs so well—just 1% worse.
>
> OSCAR 21.09 uses aggressive line-wise deduplication, which is very effective at finding duplicates. **This is one of the few dataset (along w/ C4) that is already adequately duplicated, and hence why it performs so well.**
>
> For OSCAR-22.01, the authors elected to share a public default version without deduplication, which explains the degraded performance.
>
> > Additionally, it seems that filtering/deduplication provides very small benefits for the dataset (and deduplication appears not to help C4 much at all either). Do you have any insights here?
>
> For both OSCAR 21.09 and C4, we believe that their deduplication process was already of high-quality (line-wise for OSCAR and 3-sentence span for C4). **This is visible in Appendix G Table 6, where the removal rates from deduplication on these datasets is small.** We will surface these results in the main text with the extra page of content allowed, as we believe they are of general interest to better understand data pipelines. This also opens the perspective for future work of exploring the "Pareto frontier" of deduplication: how to build deduplication methods that are cheaper and more scalable, but still match enough duplicates to retrieve most of the performance gains.
>
> Regarding the benefits of filtering, we found that these are harder to correlate with removal rates. We believe it is likely filtering is a more "finicky" process than deduplication, and requires specific tuning. Notably, depending on the text extraction used, more or less filtering may be needed to remove items such as menus, social media counters, etc.
>
>
> ## Public extract rationale
> > “We publicly release an extract of 600 billion tokens” Why only a small fraction?
>
> This is due to**commercial/internal reasons**, the management of our company only approved a partial release of the dataset and not a complete one.
>
> ## Results from larger models
> > It might be nice to include results from Falcon-40B and larger models from other groups on graphs
>
> We focused on models with comparable compute budgets to the ones trained to study and ablate RefinedWeb, to allow for apple to apple comparisons. **While Falcon-40B is trained on a significant portion of RefinedWeb, it also includes code data and multilingual data, which wouldn't make it a "fair" representative of RefinedWeb**. We note also the difficulty in finding aggregates of tasks that overlap across reported results for models, especially so for closed models (GPT-4, PaLM-2, etc.) for which only API access to a RLHF version is available, making pretrained comparisons impossible.

---

### Official Review · Reviewer_2Ywv · 2023-07-23
**Novel result showing that heaving filtering and deduplication of raw web data can produce high-quality training corpus**

**Rating:** 7
**Confidence:** 3
**Clarity:** The paper is well-written.

**Strengths:**

- The findings are interesting and the carefully filtered and deduplicated dataset that was released can greatly benefit future work.
- The data preprocessing steps are clearly explained and well-motivated. The paper also reports how much data was left after each step, as well as showing the added benefits obtained from their preprocessing steps.
- The paper compares baselines using both the same (internal) codebase as well as codebases used by previous work.

**Additional Feedback:**

N/A

**Correctness:**

The dataset is constructed in a sound way. Regarding interpretation of the results, see “Opportunities for Improvement”.

**Documentation:**

There is sufficient detail on data collection.

**Ethics:**

No. I only have minor concern with how effective the paper's attempt at NSFW filtering would be (see above).

**Limitations:**

The paper includes a study of toxicity (measured by Perspective API), and defers to future work for studying potential social biases as part of the dataset is publicly available. It would be good if the paper could use a subset of the data to do a study on potential NSFW content remaining after filtering based on URLs only.

**Opportunities For Improvement:**

- My main concern is with regards to the claim that the resulting model is equally capable as those trained on curated corpora, as the paper only reports aggregated performance. The authors should include a fine-grained breakdown and provide analysis of performance on each of the evaluation tasks. For example, does training on only Common Crawl data make the trained model less good at math and coding tasks?
-  It’d be good to include a controlled experiment where models are trained on the same number of tokens from each data source.

**Relation To Prior Work:**

Related work is thoroughly discussed.

**Summary And Contributions:**

The paper presents RefinedWeb, a 5 trillion tokens English-only dataset, obtained solely from Common Crawl. The main finding of the paper is that properly filtering and deduplicating Common Crawl data can produce language models that are as capable as those trained on curated corpora consisting of multiple high-quality sources (e.g. books, arxiv, etc.).

---

> ### Author Response · Authors · 2023-08-17
> **Answer to Reviewer 2Ywv**
>
> We would like to thank reviewer `2Ywv` for their positive assessment of our work and for their valuable review.
>
> We address below questions and concerns brought-up by the reviewer. We will also soon be submitting a revised version of the paper reflecting these comments.
>
> ## Fine-grained and domain-specific performance
>
> > The authors should include a fine-grained breakdown and provide analysis of performance on each of the evaluation tasks.
>
> **We have made a dump of our results available in a Google Sheet** (for ease of reading) here: https://docs.google.com/spreadsheets/d/1u0HqZVtNxe2bYmF_1lQneR0FH-s6TOnjiRE0bV1LtEA/edit?usp=sharing
>
> Due to the breadth of our evaluation setup, it's difficult to provide individual analysis for each task. However, on the topic of domain-specific tasks, we would like to highlight results of 1.3B parameters models on `PubMedQA` (medical domain) and `SciQ` (science questions):
>
> |                 | OpenAI-babbage | Ours-Pile | Ours-RefinedWeb |
> |-----------------|:--------------:|:---------:|:---------------:|
> | PubMedQA [acc.] |      61.2      |    60.8   |     **62.5**    |
> | SciQ [acc.]     |      86.7      |     88    |     **89.9**    |
>
> **We find that despite The Pile incorporating domain-specific medical knowledge and scientific papers, it does not outperform models trained with RefinedWeb**. We attribute this to two factors: (1) web data is broad, and likely still contain some medical and technical content, so our models are not entirely blind in these domains; (2) filtering and deduplication fundamentally improve the capabilities of models, and for these tasks, the models may benefit more from a generic improvement at Q/A than from domain specific knowledge.
>
> > For example, does training on only Common Crawl data make the trained model less good at math and coding tasks?
>
> **Our current evaluation setup is entirely focused on natural language evaluation, and does not include mathematics or coding tasks.** For these datasets, it is common to add specific datasets which can be scrapped/generated in large quantities -- similarly to web data. **We do not believe web data alone to be sufficient to train an adequate code model; code data is required.** For instance, in Falcon-40B, we added permissively licensed code data from GitHub on top of RefinedWeb to elicit code abilities.
>
> We will add a discussion in our limitations section on domain-specific tasks (code, mathematics, etc.) **Overall, the point of our paper is more to challenge the view that curated data is necessary for performant general language models (as was often brought up based on The Pile/GPT-3)**. We however agree that domain-specific data can be valuable for domain-specific tasks (especially when the domain is vastly different from natural language, as is code).
>
>
> ## Control for number of tokens
>
> > It’d be good to include a controlled experiment where models are trained on the same number of tokens from each data source.
>
> This is already the case: **all of our experiments are ran with the same pretraining duration**. So all comparisons are made with models of the same size, trained for the same number of tokens, and hence with the same compute budget. In essence, this broadly shows that one token of filtered/curated data is more valuable than one token of raw data.
>
>
> ## NSFW content
>
> > It would be good if the paper could use a subset of the data to do a study on potential NSFW content remaining after filtering based on URLs only.
>
> During development of RefinedWeb, we tried such a study but failed to obtain principled measurements. Notably, **the issue is in finding a ground truth for what constitutes NSFW content.** Many automated classification models (e.g., ToxicBERT, etc.) will report for instance LGBTQ or medical content as NSFW. Basing our filtering on these models would thus introduce a bias in our model. We ended up mostly validating our URL filter through manual inspection of reported hits, but it is difficult to report on this in a principled way.

---

### Official Review · Reviewer_GCg1 · 2023-07-27
**Valuable dataset cleaning pipeline for community**

**Rating:** 7
**Confidence:** 4
**Clarity:** Yes

**Strengths:**

1. The related works section is thorough. It covers many aspects of this work and I feel confident in the authors’ research in this area.
2. MacroData Refinement. This pipeline for filtering and deduplicating datasets has multiple steps in which to catch low-quality or duplicated data, and used on a large scale dataset not previously been done before.
3. There are several places where the authors mention how their design and pipelining decisions took bias amplification or bias mitigation into consideration which I appreciate.
4. It is valuable insight to see how model performance changes across different steps in the MacroData Refinement pipeline.
5. Evaluating models based on zero-shot performance instead of perplexity makes sense and is an interesting approach. I would have liked to see both evaluations done as there are benefits to both metrics, but I understand the authors’ reasoning.

**Additional Feedback:**

1. I don’t understand the reasoning behind removing Wikipedia and arXiv.
2. I would like more description of small-agg, core-agg, main-agg, and ext-agg.
3. Figures 1 and 3 — what is PF?
4. Perhaps outside of the scope of this paper, but I'm curious about how much the content of the dataset influences the performance across different tasks, and if by removing duplicated content, there was a decrease in performance. Do some tasks benefit more from duplicated data?

**Correctness:**

I believe the claims made in this submission are correct. The dataset was constructed well, and I agree with the design principles presented by the authors.

**Documentation:**

Dataset and models are available on HuggingFace with associated data and model cards. There is sufficient detail for reproducibility.

**Ethics:**

No ethical concerns.

**Limitations:**

The authors have fully addressed the limitations of their work.

**Opportunities For Improvement:**

1. Does the size of the dataset on affect training time? Why should researchers spend the time to filter, clean, and deduplicate large scale datasets if the performance increase is only a few percentage points and the training time is the same? How long did the refinement pipeline take on 5 trillion tokens.
2. What was the reasoning for releasing only a 600 billion token sample of the RefinedWeb dataset and not the entire dataset? How can researchers use the findings in this paper or make direct comparisons if the data is not available?
3. A lot of model performance increases are made with few-shot learning (especially for certain specialized tasks), how do model trained on RefinedWeb compare to state-of-the-art models in a few-shot setting?

**Relation To Prior Work:**

Yes.

**Summary And Contributions:**

Curated corpora is typically higher quality, but expensive to create. Massive web crawls are a cheaper way to obtain the large text sources needed for training LLMs. In this work, the authors propose a method for improving the quality of uncurated web data. They introduce a new 5 trillion token dataset, RefinedWeb, which as demonstrated improved performance on models trained on previous data sources.

---

> ### Author Response · Authors · 2023-08-17
> **Answer to Reviewer GCg1: general comments and influence of dataset quality on training time  (1)**
>
> First, we would like to thank reviewer `GCg1` for his positive assessment of our work, and for providing a thorough review.
>
> We address below questions and concerns brought-up by the reviewer; we have broken down our answer in topic-specific messages. We will also soon be submitting a revised version of the paper reflecting these comments.
>
>
> ## Dataset (and dataset quality) influence on training time
>
> > Does the size of the dataset on affect training time?
>
> Broadly speaking, the compute budget (in number of floating point operations, FLOPS) associated with training a model is $C = 6 N D$ with $N$ the number of parameters in the model and $D$ the dataset training size in tokens [1]. So training for longer (i.e., on a larger dataset) will take more time as more operations will have to be completed.
>
> Practionners typically have a fixed compute budget $C$: either derived from X GPUs being available for Y time, or from a $ budget enabling them to buy so many GPU-hours. Accordingly, **for a fixed compute budget $C$, the relevant question is often how good of a model can you train**.
>
> > Why should researchers spend the time to filter, clean, and deduplicate large scale datasets if the performance increase is only a few percentage points and the training time is the same?
>
> Based on the discussion above, we address this under the angle of how much of a performance gain can clean data give you over "bad" unfiltered data.
>
> In ablations reported in Table 3, we find that the 1.3B parameters model trained on 27B tokens of RefinedWeb outperforms a 3B parameters model trained on 60B tokens tokens on OSCAR-22.01. Accordingly, training on better data here leads to a 4x reduction in compute costs!
>
> Accordingly data filtering, cleaning, and deduplication improves performance in a way that could only be compensated by increasing training budgets significantly: not just by a few percents, but by 25-300% depending on the quality of the original dataset. This is because gaining a few percents in zero-shot aggregate performance is actually difficult, and translates significant improvements in the model. **Compute alone cannot efficiently compensate for bad data**, which motivates for the creation of high-quality datasets such as RefinedWeb.
>
> > How long did the refinement pipeline take on 5 trillion tokens?
>
> As a logical follow-up, the question is whether the filtering of the data is not more expensive than the increase in training budget that would be required to compensate for the absence of filtering.
>
> This can be a difficult question to contextualize as it will depend a lot on the underlying infrastructure available, and the end costs may dependend on negociated cloud contracts, use of spot instances, etc.
>
> However, as a ballpark, we can estimate the costs of processing the 5T tokens of RefinedWeb to be on the order of a few $100k. For the ~1T tokens used to train Falcon-40B, this represents less than 5% of the total costs of training -- and as discussed above, offsetting the degraded performance caused by cheaper data would increase GPU compute budget significantly.
>
> We note as well that the cost of data processing is amortized over many training runs: RefinedWeb was used to train many models internally, and the public extract will hopefully see even wider usage.
>
> Accordingly, and to summarize our answer to the reviewer: **training on adequately filtered and deduplicated data delivers better downstream task performance for the same pretraining compute budget, and that improvement is well worth the increased data processing costs.** It is simply more efficient to dedicate that compute/budget to data processing than to an increase in model scale.
>
> [1]: Kaplan et al., 2020. Scaling Laws for Neural Language Models

---

> ### Author Response · Authors · 2023-08-17
> **Answer to Reviewer GCg1: rationale for the partial release and few-shot results (2)**
>
> ## Rationale for the partial release of 600 billion tokens only
>
> > What was the reasoning for releasing only a 600 billion token sample of the RefinedWeb dataset and not the entire dataset?
>
> This is mostly due to **commercial reasons**; the management of our company only approved a partial release of the dataset and not a complete one.
>
> > How can researchers use the findings in this paper or make direct comparisons if the data is not available?
>
> **All results obtained within this paper were obtained with the 600GT extract** (or intermediary development versions), so they are reproducible and comparable. Furthermore, 600GT for a single-source is already significantly more than many other datasets such as The Pile, C4, or OSCAR--which should enable researchers to train interesting models.
>
> ## Few-shot learning and perplexity evaluations
>
> > A lot of model performance increases are made with few-shot learning (especially for certain specialized tasks), how do model trained on RefinedWeb compare to state-of-the-art models in a few-shot setting?
>
> **Few-shot evaluations have been found to be strongly correlated to zero-shot evaluations**: [2] found a Pearson correlation coefficient of 0.93 between zero-shot and few-shot performance for results reported in the GPT-3 paper.
>
> As few-shot evaluations are more expensive to run (longer context), more difficult to reproduce (choice of few shot samples), we have elected to focus on zero-shot evaluation only to simplify our evaluation setup. This is in line with many other works on large language models: BLOOM, GPT-3, Gopher/Chinchilla, PaLM.
>
> >  I would have liked to see both evaluations [authors: perplexity/zeros-shot] done as there are benefits to both metrics, but I understand the authors’ reasoning
>
> **We do include some perplexity evaluations in Appendix H.2. Table 8 on Wikitext**, showing that despite not seeing any data from Wikipedia, our models achieve performance close to models trained on The Pile (which explicitely include Wikipedia).
>
> During experiments, we found that such perplexity evaluation was very sensitive to formatting choices: on Wikipedia for instance, the specific layout of an article is easier to get for a model that has been explicitely trained on it. This lead us to focus our efforts on zero-shot generalisation.
>
>
> [2] Le Scao et al., 2022. What Language Model to Train if You Have One Million GPU Hours?

---

> ### Author Response · Authors · 2023-08-17
> **Answer to Reviewer GCg1: miscellaneous questions (3)**
>
> ## Miscellaneous questions
> > I don’t understand the reasoning behind removing Wikipedia and arXiv.
>
> Typically, practitioners will combine web datasets with premade dumps of specific websites (e.g., Wikipedia, arXiv, reddit, etc.) [3]. If we did not remove these, **additionnal deduplication would have to be ran when adding Wikipedia & others on top of RefinedWeb**. By removing them explicitely, this makes combining RefinedWeb with other common datasets easier.
>
> > I would like more description of small-agg, core-agg, main-agg, and ext-agg.
>
> **We detail aggregates composition in Appendix I.1. and Table 9**. These were originally in the main paper, but we had to move them due to the length constraints of NeurIPS. We are adding them back to the main text thanks to the extra page of content we have now.
>
> > Figures 1 and 3 — what is PF?
>
> **PF here stands for PetaFLOPS, a unit of compute** (1 PF-days = 1 PetaFLOPS sustained for a day). We will clarify in the updated text.
>
> > Perhaps outside of the scope of this paper, but I'm curious about how much the content of the dataset influences the performance across different tasks, and if by removing duplicated content, there was a decrease in performance. Do some tasks benefit more from duplicated data?
>
> This is indeed a rather interesting question: specific content like famous quotes, popular lyrics, etc. may be safe, and in fact even beneficial to memorize. We were unable to identify tasks which clearly exhibited a behaviour showing that deduplication was not beneficial, but these may exist outside of our evaluation setup.
>
> **We will add to the limitations section a short discussion of this.**
>
>
> [3] Gao et al., 2020. The Pile: An 800GB Dataset of Diverse Text for Language Modeling.

---

### Author Response · Authors · 2023-08-21
**Updated version of the paper following rebuttal**

First, we would like to thank all reviewers for their positive assessment (scores: 7, 7, 7, 7) and extensive reviews. Notably, `TYb3` titled his review `A step forward in the open source community's understanding and utilization of LLM pre-trained web data`, echoing `2Ywv` remarks: `the findings are interesting` and `the carefully filtered and deduplicated dataset that was released can greatly benefit future work`.  `om9q` noted that `the paper is well written and thoughtfully organized`, `full of interesting details`, and that `the evaluation is thoughtful and well-executed`,  while `GCg1` titled his review `Valuable dataset cleaning pipeline for the community` and noted `I feel confident in the authors’ research in this area`.

We have addressed questions and concerns from reviewers directly with individual answers, and **we outline below some of the key steps we have taken to further improve our paper thanks to feedback from the reviewers.** We have updated the revised version for reviewers to validate.

| **Concern/question** | **Reviewers** | **Mitigation** | **Changes** |
|------------------------------------------------------|------------------------|----------------------------------------------------------------------------------------------------------------------------------------------------------------------------------------------------------------------------------------------------------------------------------------------------------------------------------------------------------------------------------------------------------------------------------------------------------------------------------------------------------------------------------------------------|----------------------------------------------------------------------------------------------------------|
| **Domain-specific performance** | `GCg1`, `2Ywv`, `om9q` | We have added paragraphs to our limitations section regarding performance beyond natural language and beyond pretraining. **We do not believe that web data alone can produce good code models**; however, many of our findings on deduplication equally apply to code datasets [1]. Furthermore, we find that models trained on RefinedWeb outperform models trained on The Pile even on some domain-specific tasks (namely PubMedQA--The Pile includes PubMed), highlighting that **better natural language capabilities can still bring broad improvements.** | Page 10, Section 5, l299-l314. Page 9, Section 4.2, l269-270. |
| **Influence of dataset quality on pretraining time** | `GCg1`, `om9q` | We have added a short discussion showcasing that **dataset selection can account for a 4x difference in pretraining compute**, as evidenced by our ablations in Table 4. Pretraining compute alone cannot efficiently compensate for degraded data quality. | Page 8, Section 4.2, l249-251. |
| **Comments on the performance of OSCAR/C4** | `om9q`, `TYb3` | We have surfaced in Section 4.3 results previously in the Appendix, regarding applying our data pipeline to other datasets. **We find that OSCAR-21.09 and C4 are already relatively well deduplicated, while OSCAR-22.01 is distributed by default without any deduplication, explaining its position as an outlier in performance.** | Page 9, Section 4.3, l270-l292. |
| **Details of the evaluation setup** | `GCg1`, `2Ywv` | We have moved Table 3 (detailing our aggregates) from the Appendix to the main text. We have also clarified that **all our internal experiments are ran for the same amount of tokens across datasets, and that we run a control experiment on The Pile when scaling-up to control for the influence of our pretraining setup**. We also added **detailed results on all tasks evaluated in a separate spreadsheet**. | Page 7, Table 4. Page 8, Section 4.2, l240-241 and l253-254. Page 29, Appendix H.1, l835-836. |
| **Clarifications** | `GCg1`, `om9q` | We have explained the PF-days acronym, and added the motivation for the partial release to the data sheet. We have also moved figures around in the Appendix to make it more pleasant to read, which should better surface some of our discussion on perplexity evaluations (Page 28, Appendix G.2., Table 9) and our decision to explicitly remove common curated sources for RefinedWeb (Page 34, Appendix I.1.3., l913-920). | Page 1, Figure 1. Page 22, Appendix C, Table 4. Page 19-40, Appendix. |


[1]: The Stack: 3 TB of permissively licensed source code. Kocetkov, et al. 2022.

---

### Decision · Program_Chairs · 2023-09-22

**Decision:**

Accept (Poster)

**Comment:**

This paper produces a large dataset from common crawl, and via careful cleaning, demonstrates that trained models are very capable.  The results in this paper will be interesting to the open-source community and could also provide insights on data curation and cleaning more broadly.